# Temperature regulates synaptic subcellular specificity mediated by inhibitory glutamate signaling

**Mengqing Wang**[1], **Daniel Witvliet**[2,3], **Mengting Wu**[1], **Lijun Kang**[4], **Zhiyong Shao**[1]*

**1** Department of Neurosurgery, State Key Laboratory of Medical Neurobiology and MOE Frontiers Center for Brain Science, Institutes of Brain Science, Zhongshan Hospital, Fudan University, Shanghai, China, **2** Lunenfeld-Tanenbaum Research Institute, Mount Sinai Hospital, Toronto, Ontario, Canada, **3** Department of Molecular Genetics, University of Toronto, Toronto, Ontario, Canada, **4** Department of Neurobiology and Department of Neurosurgery of the First Affiliated Hospital, Zhejiang University School of Medicine, Hangzhou, Zhejiang, China

* shaozy@fudan.edu.cn

**Data Availability Statement:** All data generated in this study are submitted either in main or supplemental data set.

**Funding:** This research was supported by Natural Science Foundation of China (31872762),

## Abstract

Environmental factors such as temperature affect neuronal activity and development. However, it remains unknown whether and how they affect synaptic subcellular specificity. Here, using the nematode *Caenorhabditis elegans* AIY interneurons as a model, we found that high cultivation temperature robustly induces defects in synaptic subcellular specificity through glutamatergic neurotransmission. Furthermore, we determined that the functional glutamate is mainly released by the ASH sensory neurons and sensed by two conserved inhibitory glutamate-gated chloride channels GLC-3 and GLC-4 in AIY. Our work not only presents a novel neurotransmission-dependent mechanism underlying the synaptic subcellular specificity, but also provides a potential mechanistic insight into high-temperature-induced neurological defects.

## Author summary

Environmental temperature affects neuronal development and functions. However, it is largely unknown whether and how the temperature affects the neurodevelopment, specifically at the level of synaptic specificity. In this study, we found that high cultivation temperature results in the deficits in synaptic specificity. The high temperature induced synaptic defect requires the conserved vesicular glutamate transporter EAT-4 and the inhibitory glutamate gated chloride channels GLC-3 and GLC-4 receptors. These findings uncover a critical role of glutamatergic transmission in regulating synaptic specificity, and provide potential pathological insights into the high temperature related neurological disorders.

## Introduction

Normal brain functions require precise synaptic connectivity among billions of neuronal and non-neuronal cells. Synaptic targeting happens not only at the cellular, but also at the

Shanghai Municipal Science and Technology Major Project (No. 2018SHZDZX01) and ZJLab to ZS. The funders had no role in study design, data collection and analysis, decision to publish, or preparation of the manuscript.

**Competing interests:** The authors have declared that no competing interests exist.

subcellular level [1–3]. For example, in mouse cerebellum, basket neurons specifically form synapses at the axon initial segment of purkinje neurons [4]. Similarly, *C. elegans* specific AIY presynaptic region targets onto the RIA interneurons [5,6]. In the last couple of decades, studies have identified many genetic factors required for synaptic subcellular specificity, including secreted and adhesion molecules [4,6–16]. Additionally, synaptic development is also regulated by neural activity [17–19]. However, it is largely unknown whether environmental-dependent neuronal activity is involved in the synaptic subcellular specificity.

Temperature is a special environmental factor that can affect neuronal development and functions through activity-dependent manner [20–25]. Neuronal activity plays critical roles in neural circuitry development [18,19]. In vertebrates, neuronal activity is essential for synapse formation in the visual system [26–28]. In invertebrates, neural circuitry was traditionally thought to be hardwired and regulated by activity-independent mechanisms [29–34]. However, recent studies show that neural activity is involved in the circuit development and remodeling in *Drosophila* [35–38]. Similarly, in *C. elegans*, neuronal activity can modulate neurite growth and branching [39–42], cell fate determination [43], presynaptic remodeling and dendritic spine density [44,45]. However, it is unknown whether and how temperature or neuronal activity affects the synaptic subcellular specificity.

The nematode *C. elegans* AIY interneurons are part of the thermotaxis circuit [46–51]. In this circuit, sensory neurons such as AFD and AWC sense the thermal information and transmit it to the AIY interneurons through glutamatergic synapses [46,51–54]. The information is further passed from AIY to the next layer interneurons including RIA and AIZ [46,51]. Although the thermotaxis circuit is known for a long time, the detailed circuit connectivity is not completely understood, and the regulatory mechanisms underlying the circuit formation are largely unknown.

AIY forms stereotypic presynaptic distribution [5,6]. With this system, we previously found that the epithelial CIMA-1, a sialic acid transmembrane transporter, is required for maintaining the subcellular specificity of the AIY interneurons. In *cima-1* loss-of-function mutants, ectopic synapses emerge in the AIY asynaptic region partially due to the posterior displacement of ventral cephalic sheath cells (VCSC) glial endfeet [55]. However, ablating the VCSC glia did not completely suppress the *cima-1* ectopic synapses, suggesting that additional signals, most likely from the nervous system, are involved [55].

In this study, we showed that the AIY ectopic synaptic formation in *cima-1* loss-of-function mutants requires the inhibitory glutamate signaling, which is mediated by the ASH expressed vesicular glutamate transporter EAT-4 and the AIY expressed pLGIC family glutamate gated chloride channels GLC-3 and GLC-4. Additionally, we showed that wild-type animals cultivated at high temperature display ectopic AIY presynaptic phenotype mimicking the *cima-1* mutants. The glutamate transporter EAT-4 in ASH and the glutamate gated chloride channels GLC-3 and GLC-4 in AIY are required for both *cima-1* and high-temperature-induced ectopic synapse formation in AIY neurons. Our study not only uncovers a novel role of the glutamatergic transmission in synaptic subcellular specificity, but also provides potential pathological insights into the high temperature-induced neurodevelopmental defects.

## Results

### Glutamatergic neurotransmission regulates the AIY presynaptic subcellular specificity

The *C. elegans* AIY neurons are a pair of bilaterally symmetric neurons in the head with stereotypical synaptic distribution: the ventral asynaptic zone 1 region, the synaptic-enriched zone 2 region, and the distal synaptic-scattered zone 3 region [5,6] (Fig 1A). The sialin homolog

CIMA-1 in epidermal cells and the ADAMTS metalloprotease MIG-17 in muscles are required to maintain AIY presynaptic subcellular specificity mediated by the VCSC glia morphology during adult stage [55,56]. Incomplete suppression of the *cima-1(wy84)* ectopic synapses by VCSC glia ablation implies that neuronal signaling is involved in the synaptic subcellular specificity (see the model in Fig 1A and [55]).

Neuronal activity plays important roles in circuit formation [18,19]. To determine if the ectopic AIY presynaptic phenotype requires neuronal activity, we used synaptic transmission defects *unc-13(e1091)* mutants [57]. We found that *unc-13(e1091)* mutants displayed normal AIY presynaptic distribution (S1A–S1C and S1J Fig), consistent with previous findings that synaptic transmission is not required for normal synaptic formation [31,33,34].

Next, we asked if neurotransmission was required for the ectopic synaptic formation in *cima-1(wy84)* mutants. To address the question, we made *cima-1(wy84);unc-13(e1091)* double mutants, and found that *unc-13(e1091)* robustly suppressed the ectopic synapses in *cima-1 (wy84)* mutants (90.19% of animals displayed ectopic synapses in *cima-1(wy84)* vs 23.38% in *cima-1(wy84);unc-13(e1091)* mutants, p<0.0001, Fig 1C–1E and 1O). Those data indicate that neurotransmission is required for the ectopic synapse formation in the *cima-1(wy84)* mutants.

To determine which type of neurotransmission is required, we blocked the glutamatergic, GABAergic, cholinergic, or dopaminergic neurotransmission via the following loss-of-function mutants: *eat-4*, *unc-47*, *unc-17* and *cat-2*, which encode the vesicle glutamate transporter, vesicle gamma-aminobutyric acid γ (GABA) transporter, acetylcholine transmembrane transporter and dopamine biosynthetic enzyme respectively [58–61]. Consistent with that seen in *unc-13(e1091)* mutants, the AIY synaptic distribution was normal in those single mutants (S1C–S1J Fig), suggesting that those types of neurotransmission are not required for synaptic spatial specificity per se. Then, we tested their roles in the ectopic synaptic formation in *cima-1 (wy84)* mutants. We found that the ectopic synapses were robustly suppressed only by *eat-4 (ky5)* as assayed with both synaptic vesicle marker GFP::RAB-3 (90.19% of animals with ectopic synapses in *cima-1(wy84)* and 20.98% in *cima-1(wy84);eat-4(ky5)*, p<0.0001. Fig 1F and 1O), and the synaptic active zone marker GFP::SYD-1 (Fig 1L–1O), but not by *unc-47 (n2409)*, *unc-17(cn355)*, or *cat-2(e1112)* mutations (S2A–S2E Fig). The effect of *eat-4(ky5)* on suppressing the *cima-1(wy84)* ectopic synapses was validated with two additional loss-of-function *eat-4(nj2)* and *eat-4(nj6)* alleles [51] (Fig 1G, 1H and 1O). To exclude the possibility that the suppression of the *cima-1* ectopic synapses is *wy84* allele-specific, we tested another independently isolated *cima-1* allele *gk902655* that harbors a nonsense mutation at the R476 site [55,62]. Consistent with the *cima-1(wy84)* data, we found that both *unc-13(e1091)* and *eat-4 (ky5)* suppressed the AIY presynaptic specificity defects induced by *cima-1(gk902655)* (72.65%, 9.37% and 12.11% of animals displayed ectopic synapses in *cima-1(gk902655)*, *cima-1 (gk902655);unc-13(e1091)* and *cima-1(gk902655);eat-4(ky5)* respectively, p<0.0001 for both comparison, Fig 1I–1K and 1O). These data suggest that the suppression of the ectopic synapses is not *wy84* allele specific.

To further confirm the requirement of *eat-4* for the AIY synaptic phenotype in *cima-1* mutants, we quantified the expressivity of the ectopic synapses by measuring the ventral synaptic length and the ratio of the ventral to total synaptic length. In the *cima-1(wy84)* mutants, the ventral synaptic length and the ratio of the ventral to total synaptic length increased dramatically due to the formation of ectopic synapses (the length and the ratio are 8.65μm and 0.21 in WT; vs 16.68μm and 0.33 in the *cima-1(wy84)* mutants, P<0.0001. Fig 1M and 1N). Consistent with the penetrance data described above, both the ventral synaptic length and the ratio in *cima-1(wy84)* mutants were significantly suppressed by *eat-4(ky5)* (the length and the ratio are 16.68μm and 0.33 in *cima-1(wy84)*; vs 9.38μm and 0.23 in the *cima-1(wy84);eat-4(ky5)* double mutants, P<0.0001. Fig 1P and 1Q).

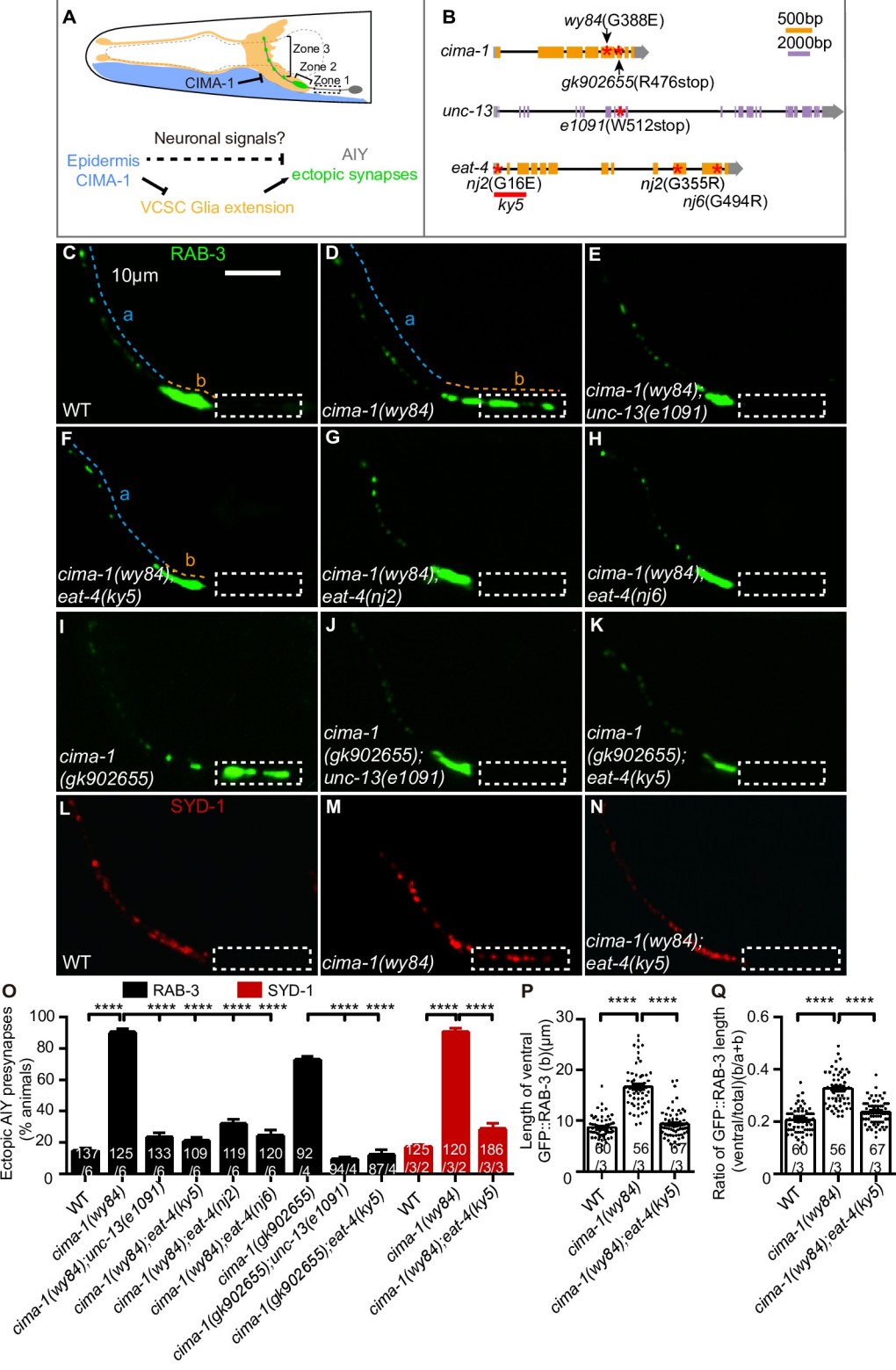

**Fig 1. Glutamatergic neurotransmission is required for the AIY ectopic synaptic formation in *cima-1(wy84)*. (A)** A model of *cima-1* in epidermal cells (blue) regulating AIY (gray) synaptic position (green) partially through modulating VCSC glia (yellow) morphology. The AIY presynaptic pattern is stereotypic and can be subdivided into three typical zones: the ventral asynaptic zone 1 region (dashed box), the synaptic enriched zone 2 region (skewed bracket), and the

distal synaptic sparse zone 3 region (vertical bracket) [5,6,55]. CIMA-1 regulates the AIY presynaptic subcellular specificity only partially mediated by the VCSC glia, suggesting that neuronal signaling is involved in the pathway. **(B)** Diagrams of the *cima-1*, *unc-13* and *eat-4* genomic structures, respectively. Exons and introns are indicated by boxes (purple or yellow boxes are translated regions; gray boxes are untranslated regions) and black lines. Mutant sites are marked with red asterisks or underlines. The purple scale bar is 2kb, and the yellow is 500bp. **(C-N)** Representative confocal micrographs of the AIY synaptic vesicle marker GFP::RAB-3 (C-K) or active zone marker SYD-1::GFP (pseudo-red, L-N) in wild-type (C, L), *cima-1(wy84)* (D, M), *cima-1(wy84);unc-13(e1091)* (E), *cima-1(wy84);eat-4(ky5)* (F, N), *cima-1(wy84);eat-4(nj2)* (G), *cima-1(wy84);eat-4(nj6)* (H), *cima-1(gk902655)* (I), *cima-1(gk902655);unc-13(e1091)* (J)and *cima-1(gk902655);eat-4(ky5)* (K) mutant adult animals. The dashed boxes indicate the zone 1 region. The scale bar in (C) is 10μm, applying to (D-N). **(O-Q)** Quantification of the AIY presynaptic pattern. Quantification of the percentage of animals with the ectopic AIY synaptic vesicle GFP::RAB-3 (black bars) and active zone GFP::SYD-1 (red bars) (L), the ventral presynaptic length (b indicated in C, D, or F) based on GFP::RAB-3 (P), and the ratio of the ventral to the total presynaptic length (b/(a+b)) based on GFP::RAB-3 (Q). In the graph, the total number of independent animals (N) and the number of biological replicates (n1) are indicated in each bar for each genotype as N/n1. And for the transgenic lines created, the number of independent transgenic lines (n2) examined, which were indicated in each bar for each genotype as N/n1/n2. For P and Q, each spot represents the value from a single AIY of a worm. Statistics are based on one-way ANOVA with Dunnett's test. Error bars are SEM. n.s., not significant, ∗∗∗∗P< 0.0001.

Collectively, these data indicate that glutamatergic neurotransmission is required for the ectopic synaptic formation in *cima-1(wy84)* mutants.

## *eat-4* acts in the ASH neurons to regulate the AIY synaptic subcellular specificity

To understand where *eat-4* acts to regulate the AIY synaptic subcellular specificity, we performed tissue-specific rescue by expressing *eat-4* cDNA in different tissues or cell types. We found that *eat-4* completely rescued and restored the ectopic synapses in *cima-1(wy84);eat-4(ky5)* double mutants when expressed in the nervous system with *rab*-3 promoter [63], or in the glutamatergic neurons with *eat-4a* promoter (S3A Fig, [64]), but not in the VCSC glia, epidermis, muscle, intestine or AIY interneurons with *hlh-17*, *dpy-7*, *myo-3*, *ges-1* and *ttx-3* promoters respectively [65–69] (Fig 2A and 2B). The data further support the hypothesis that glutamatergic neurotransmission is required for the ectopic synapse formation in *cima-1 (wy84)* mutants.

To further determine the specific glutamatergic neuron(s) involved in the AIY ectopic synaptic formation in *cima-1(wy84)* mutants, we expressed *eat-4* cDNA in the glutamatergic neurons previously identified as AIY synaptic partners including the presynaptic AUA (P*flp-8*) [70], ASE (P*gcy-5*)[71], AFD (P*gcy-8*)[71], AWC (P*str-2*)[72], ASG and BAG (P*eat-4b*: 4454bp to 3554bp upstream regulatory sequence) [64] and the postsynaptic RIA (P*glr-3*) neurons [5,73]. To our surprise, none of them rescued (Fig 2A and 2B). Then, we expressed *eat-4* in twelve pairs of sensory neurons, including AWC, ASG, ASH, ASK, ADL, PHA and PHB seven pairs of glutamatergic neurons with *odr-4* promoter [74]. Interestingly, this transgene fully rescued (Fig 2A and 2B). Finally, we tested the rescue in ASH, ASK or ADL with *nhr-79* (or *sra-6*), *sra-9* and *srh-220* promoter respectively [75–78], but not in others because AWC and ASG were excluded previously and PHA and PHB are located in the tail, far away from AIY neurons. Interestingly, robust rescue was observed when *eat-4* was expressed in the ASH, and to a less degree in ASK, but not in ADL neurons (Fig 2A and 2B). The data suggest that *eat-4* acts mainly in the ASH to promote the AIY ectopic synapse formation in *cima-1(wy84)* mutants.

VGLUT overexpression leads to increasing glutamate loading in the synaptic vesicle and enhancing glutamate release in *Drosophila* [79,80] and vertebrates [81,82]. To determine whether overexpression of EAT-4/VGLUT is sufficient to induce the AIY ectopic synapses, we overexpressed the *eat-4* cDNA in the ASH with different promoters, and found that they all robustly induced the ectopic synapses (Fig 2C). The data suggest that *eat-4(OE)* in the ASH is sufficient to induce the AIY ectopic synapse formation.

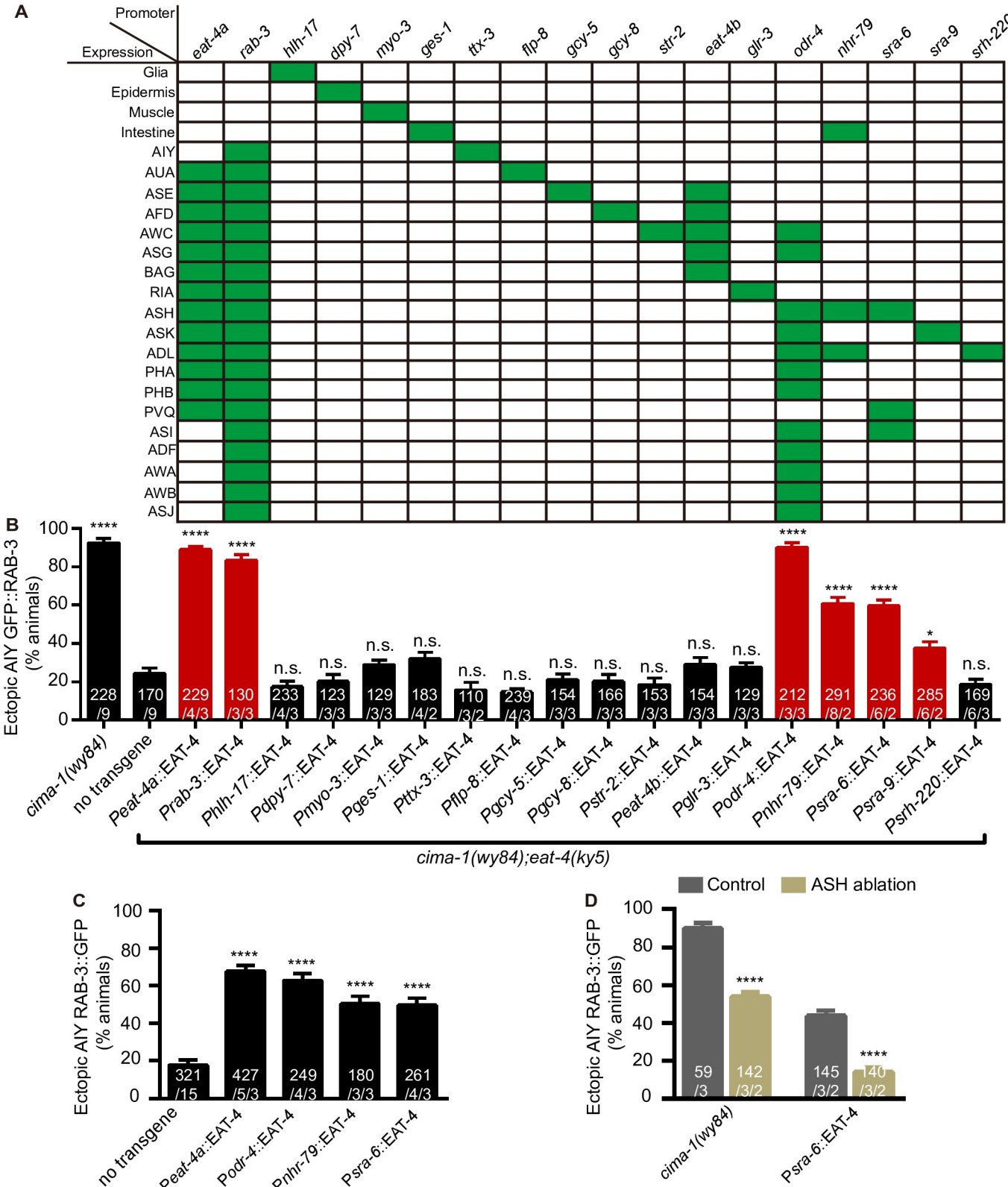

**Fig 2. *eat-4* acts mainly in the ASH to regulate the AIY synaptic subcellular specificity. (A)** The tested tissue-specific promoters (first row) and the tissues/neurons were listed in the table. Green boxes indicate the expressing tissues/neurons, while the empty boxes indicate the non-expressing ones. Note that the neurons expressing *eat-4* cDNA through *eat-4a* or *rab-3* promoter include but are not limited to those listed in the table. **(B-D)** Quantification of the percentage

of animals with the ectopic AIY synaptic GFP::RAB-3 in the zone 1 region for tissue-specific rescue (B), tissue-specific overexpression (C) and ASH ablation (D) for the indicated genotypes. The data in (B) collectively demonstrate that *eat-4* expressed in the ASH neurons contributes to the major portion of the animals with the ectopic synapses. The data in (C) show that *eat-4* overexpression in the ASH is sufficient to induce the ectopic synapses in the AIY zone 1 region. The data in (D) showed that ASH is required for the ectopic synaptic formation in *cima-1(wy84)* or ASH-specific *eat-4(OE)* (P*sra-6*::EAT-4) animals. Error bars are SEM. *P< 0.05, ****P< 0.0001, n.s., not significant. Statistics are based on one-way ANOVA with Dunnett's test (B, C) or unpaired t test (D). The total number of independent animals (N) and the number of biological replicates (n1) are indicated in each bar for each genotype, as are, for the transgenic lines created, the number of independent transgenic lines (n2) examined (using the convention N/n1 or N/n1/n2).

To further confirm the role of ASH neurons in regulating AIY synaptic specificity, we ablated the ASH neurons by expressing apoptotic protein caspase-3 [83]. We observed that ASH ablation partially but significantly suppressed the ectopic synapses induced by *cima-1 (wy84)* (89.91% and 53.94% of animals with ectopic synapses in ASH-normal and -ablated animals respectively, p<0.0001. Fig 2D), and completely abolished the ectopic synapses induced by *eat-4* overexpression in the ASH (43.97% and 14.09% of animals with ectopic synapses in ASH-normal and -ablated animals respectively, p<0.0001. Fig 2D). Those data further support that the glutamate required for the AIY ectopic synaptic formation is mainly from the ASH sensory neurons.

CIMA-1 regulates the AIY synaptic position mediated partially through VCSC glia [55,56]. To address if the VCSC glia is required for the glutamatergic signaling induced ectopic synapse formation, we ablated the VCSC glia in wild-type, *cima-1(wy84)* and *eat-4(OE)* animals. In wild-type animals, loss of the glia did not affect synaptic distribution (18.31% and 15.75% of total animals with ectopic synapses in wild type and glia-ablated animals respectively, p = 0.4788. S4A Fig). In *cima-1(wy84)* mutants, glia ablation partially suppressed the ectopic synaptic distribution (93.11% and 66.08% of total animals with ectopic synapses in glia-normal and -ablated animals respectively, p = 0.0023. S4A Fig), which is consistent with previous studies [55]. Interestingly, in *eat-4(OE)* (P*eat-4a*::EAT-4 transgenic) animals, glia ablation only slightly suppressed the ectopic synapses (62.19% and 53.37% of total animals with ectopic synapses in glia normal and ablated animals, p = 0.0047. S4A Fig). The data indicated that VCSC glia only contribute a little to the synaptic defect induced by *eat-4(OE)*. In other words, *eat-4 (OE)* promotes the AIY ectopic synaptic formation largely in a glia-independent manner.

To address when *eat-4* acts, we quantified the AIY synaptic distribution at different developmental stages in *eat-4(OE)* animals. Interestingly, the ectopic synapses in *eat-4(OE)* emerged since the larval L1 stage (S5A–S5I Fig), unlike in *cima-1(wy84)* mutants which shows up only at adult stage [55]. Consistently, we found that the ventral synaptic length and the ratio of ventral to total synaptic length were significantly increased since the L1 stage (S5H–S5I Fig). Furthermore, the *eat-4* embryonic expression supports its early role in AIY synaptic subcellular specificity (S3B and S3B' Fig). These data collectively indicate that *eat-4(OE)* and *cima-1 (wy84)* may promote the AIY ectopic synaptic formation through different mechanisms.

The synapses in zone 2 of wild-type animals are formed primarily onto the postsynaptic partner RIA [5]. To determine whether the ectopic synapses in *eat-4(OE)* are targeted to RIA, we simultaneously labeled RIA neurons and the AIY presynaptic sites, and found that the AIY ectopic presynaptic sites were only partially in apposition to the RIA neurons (S5J and S5K Fig), suggesting that some of the AIY ectopic synapses are not targeting onto RIA.

## Glutamate-gated chloride channels GLC-3 and GLC-4 mediate the ectopic synapse formation

To address which glutamate receptor(s) is required, we analyzed all four types of glutamate receptors that have been identified in *C. elegans* including AMPA receptors, NMDA receptors, metabotropic G-protein-coupled receptors and glutamate-gated chloride channels (GluCls)

(Fig 3A) [84–86]. Among eighteen loss-of-function receptors we tested, all of them displayed the normal AIY synaptic subcellular distribution (S6A–S6R Fig), suggesting that those receptors are not required for the AIY presynaptic subcellular specificity per se, which is consistent with the *eat-4* loss-of-function phenotype seen above.

Then, we tested the roles of those receptors in suppressing *cima-1(wy84)* mutant phenotype. Interestingly, two glutamate-gated chloride channel mutants, *glc-3(ok321)* and *glc-4 (ok212)* partially but significantly suppressed the *cima-1(wy84)* ectopic synapses formation as assayed with the synaptic vesicle marker GFP::RAB-3 (89.91% of animals with ectopic synapses in *cima-1(wy84)*; 53.91% in *cima-1(wy84);glc-3(ok321)*; 67.57% in *cima-1(wy84);glc-4(ok212)*, p<0.0001 and p = 0.0029 as compared to *cima-1(wy84)* respectively. S7A–S7U Fig), while the rest mutant receptors did not. And *glc-3(ok321);glc-4(ok212)* double mutations completely suppressed the ectopic synapses in *cima-1(wy84)* mutations (89.84% of animals with ectopic synapses in *cima-1(wy84)*, 19.13% in *cima-1(wy84);glc-3(ok321);glc-4(ok212)*, p<0.0001. Fig 3B, 3C and 3F). The suppression effect by *glc-3(ok321);glc-4(ok212)* was confirmed with the active zone marker GFP::SYD-1 (89.66% of animals with ectopic synapses in *cima-1(wy84)*; 28.76% in *cima-1(wy84);glc-3(ok321);glc-4(ok212)*, p<0.0001. Fig 3D–3F). Consistently, both the ventral synaptic length and the ratio of ventral to total synaptic length in *cima-1(wy84)* mutants were robustly suppressed by *glc-3(ok321);glc-4(ok212)* double mutations (the length and the ratio are 16.76μm and 0.34 in *cima-1(wy84)*; vs 9.77μm and 0.22 in *cima-1(wy84);glc-3 (ok321);glc-4(ok212)* mutants, P<0.0001 as compared to *cima-1(wy84)*. Fig 3G and 3H). The role of *glc-3(ok321)* and *glc-4(ok212)* in suppressing *cima-1* was confirmed by *cima-1 (gk902655)* allele (S7V–S7Z Fig). Together, the data suggest that the ectopic synapse formation in *cima-1* mutants requires the glutamate-gated chloride channels GLC-3 and GLC-4.

Next, we tested whether the ectopic synapses induced by *eat-4(OE)* also requires GLC-3 and GLC-4. We found that either *glc-3(ok321)* or *glc-4(ok212)* partially suppressed the *eat-4 (OE)*-induced ectopic synapses (18.07% of animals with ectopic synapses in wild type; 66.46% in *eat-4(OE)*, p<0.0001 as compared to wild type; 45.98% in *eat-4(OE);glc-3(ok321)*, p = 0.0061 as compared to *eat-4(OE)*; 44.38% in *eat-4(OE);glc-4(ok212)*, p = 0.0032 as compared to *eat-4 (OE)*. Fig 3I–3L and 3N). Notably, *glc-3(ok321);glc-4(ok212)* double mutations completely suppressed the ectopic synaptic formation induced by *eat-4(OE)* (22.95% of animals with ectopic synapses in *eat-4(OE);glc-3(ok321);glc-4(ok212)*, p<0.0001 as compared to *eat-4(OE)*. Fig 3J, 3M and 3N). These data collectively suggest that *eat-4(OE)* promotes the AIY ectopic synaptic formation through the glutamate-gated chloride channels GLC-3 and GLC-4.

## GLC-3 and GLC-4 act cell-autonomously in AIY to promote the ectopic synaptic formation

To understand where GLC-3 and GLC-4 act to promote the AIY ectopic synapse formation, we firstly determined where they were expressed by generating transcriptional reporter P*glc-3*:: GFP and P*glc-4*::GFP, co-labeled with the AIY reporter P*ttx-3*::mCherry [69]. We found that both P*glc-3*::GFP and P*glc-4*::GFP were expressed in head neurons including the AIY (Fig 4A–4B"). We also noticed that both *glc-3* and *glc-4* were expressed since the embryo stage (S3C-S3D' Fig), which is consistent with their role in mediating the ectopic synapse formation of *eat-4(OE)* animals at the L1 stage (S5A–S5I Fig). Next, we performed cell-specific rescue by driving *glc-3* or *glc-4* cDNA with AIY specific (*ttx-3*) promoter [69], with endogenous promoters as controls. We found that expressing *glc-3* or *glc-4* with AIY specific *ttx-3* promoter rescued the corresponding mutants to the degree as with the endogenous promoters (Fig 4C). Additionally, we found that overexpressing *glc-3* and *glc-4* simultaneously in the AIY of wild-type animals induced the ectopic synapses in *eat-4*-dependent manner (Fig 4D–4F and 4H).

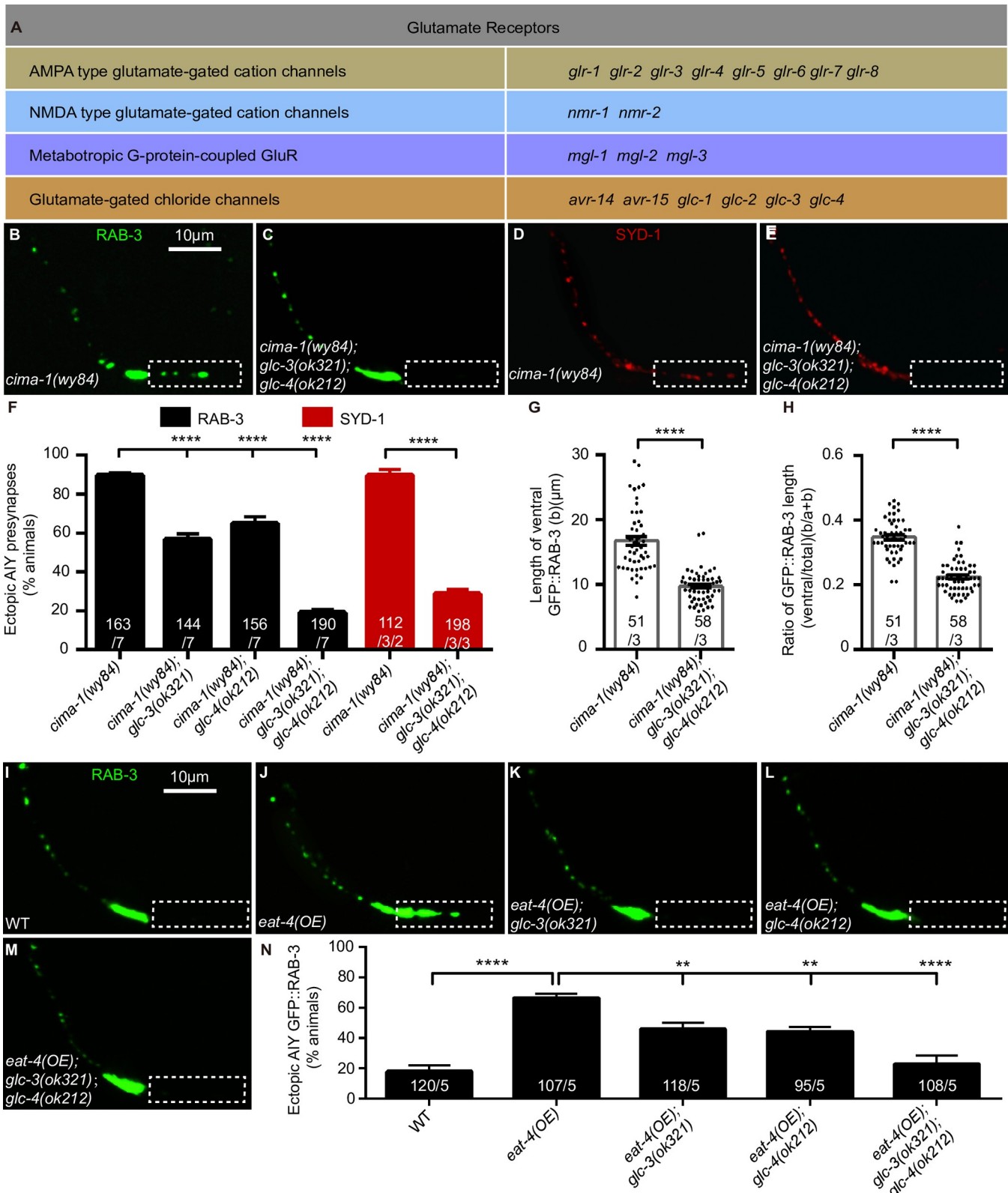

**Fig 3. Glutamate-gated chloride channels GLC-3 and GLC-4 are required for the ectopic synaptic formation. (A)** A list of genes encoding four type of glutamate receptors tested for the role in the ectopic synaptic formation: AMPA receptors, NMDA receptors, metabotropic glutamate receptors and glutamate-gated chloride channels. **(B-E)** Representative confocal micrographs of the AIY synaptic vesicle marker GFP::RAB-3 (B and C) or active zone marker SYD-1::

GFP (pseudo-red, D and E) in *cima-1(wy84)* (B and D), *cima-1(wy84);glc-3(ok321);glc-4(ok212)* (C and E). The dashed boxes indicate the zone 1 region. The scale bar in (B) is 10μm, applying to (C-E). **(F)** Quantification of the percentage of animals with the ectopic synapses in the AIY zone 1 region for indicated genotypes. Either *glc-3(ok321)* or *glc-4(ok212)* partially suppresses the ectopic synapses in *cima-1(wy84)*, and the *glc-3(ok321);glc-4(ok212)* double mutations enhance each single mutation and suppress to the degree as *eat-4(ky5)* does. **(G and H)** Quantification of the ventral presynaptic length (G) and the ratio of the ventral to the total presynaptic length (H) based on the GFP::RAB-3 marker. **(I-M)** Representative confocal micrographs of the AIY presynaptic marker GFP::RAB-3 in wild-type (I), *eat-4* overexpression (*eat-4(OE)*) (J), *eat-4(OE);glc-3(ok321)* (K), *eat-4(OE);glc-4(ok212)* (L) and *eat-4(OE);glc-3(ok321);glc-4(ok212)* (M) animals. The dashed boxes indicate the zone 1 region. The scale bar in (I) is 10μm, applying to (J-M). **(N)** Quantification of the percentage of animals with the ectopic synapses in the AIY zone 1 for indicated genotypes. Either *glc-3(ok321)* or *glc-4(ok212)* single mutation partially suppresses, while the *glc-3(ok321); glc-4(ok212)* double mutations completely abolish the ectopic AIY presynaptic distribution induced by *eat-4(OE)*, indicating that the ectopic synapses induced by glutamate over-release is GLC-3- and GLC-4-dependent. For (F-H) and (N), the total number of independent animals (N) and the number of biological replicates (n1) are indicated in each bar for each genotype. And for the transgenic lines created in (F-H), the number of independent transgenic lines (n2) examined is indicated as the convention N/n1/n2. For (N), the transgene (*eat-4(OE)*) in these genotypes is from the same one transgenic line. Statistics are based on one-way ANOVA with Dunnett's test (N and black columns in F) or unpaired t test (G, H and red columns in F). Error bars are SEM. $^{**}P < 0.01$, $^{****}P < 0.0001$.

Those data reveal that two glutamate-gated chloride channels GLC-3 and GLC-4 act cell-autonomously in AIY to modulate the synaptic subcellular specificity.

Given that GLC-3 and GLC-4 mediate inhibitory neurotransmission [51,87,88], we speculated that they induced the ectopic synapses through inhibiting AIY activity. To test this possibility, we expressed the gain-of-function potassium channel UNC-103(A334T) in AIY neurons. The gain of function UNC-103(A334T) can inhibit neuron excitability [89–92]. Indeed, the AIY-specific *unc-103(gf)* expression resulted in the AIY ectopic presynaptic formation in the zone 1 (Fig 4G and 4H), supporting the model that inhibiting the AIY activity is sufficient to induce the ectopic presynaptic assembly.

To directly examine if ASH-specific *eat-4(OE)* affects AIY activity, we recorded the calcium signaling in AIY with GCaMP6s[93]. We found while the amplitude of the automatic calcium oscillation was not affected, the frequency was dramatically reduced (Fig 4I–4K and S1 Video). These results support the model that the glutamate transmission from ASH promotes the AIY ectopic synaptic assembly through inhibiting its activity.

To further understand how GLC-3 and GLC-4 regulate AIY synaptic specificity, we determined GLC-3 and GLC-4 localization in AIY with AIY-specific mCherry::GLC-3 and mCherry::GLC-4 reporters. Interestingly, both GLC-3 and GLC-4 clusters largely overlapped with the synaptic marker GFP::RAB-3 in the zone 2 in wild-type or *eat-4(ky5)* mutants, and they were not present at the zone 1 region (Fig 5A–5B' and 5F–5G'). In *cima-1(wy84)* or the ASH-specific *eat-4(OE)* animals, however, the GLC-3 and GLC-4 were also ectopically colocalized with the GFP::RAB-3 in the zone 1 region (Fig 5C, 5C', 5E, 5E', 5H, 5H', 5J and 5J'). Loss of *eat-4* suppressed the *cima-1(wy84)*-induced ectopic distribution of GLC-3 and GLC-4 as well as GFP::RAB-3 in the zone 1 (Fig 5D, 5D', 5I and 5I'), suggesting that GLC-3 and GLC-4 probably act locally to promote presynaptic assembly.

## ASH neurons are AIY presynaptic partners

Our above results demonstrate that overexpressing *eat-4* specifically in ASH neurons promotes the AIY ectopic presynaptic formation through inhibiting its activity mediated by GLC-3/GLC-4 receptors. Those data implied that the ASH neurons most likely form synapses onto AIY, which was not reported previously [5,94]. To test this hypothesis, we examined electron microscopy (EM) reconstructions of three hermaphrodite nerve rings [5,95]. Interestingly, we found that ASH formed a chemical synapse onto AIY on one or both of the left-right pairs at the anterior region of zone 1, where the ectopic synapses begin to form in *cima-1(wy84)* or *eat-4(OE)* animals (Fig 6A and S2 Video) [5, 95]. And we found that the ASH neurons were extended posteriorly and aligned next to the AIY zone 1 in *cima-1(wy84)* mutants (S8A-S8B" Fig), which makes it possible for ASH to form extra synapses onto the AIY in the zone 1.

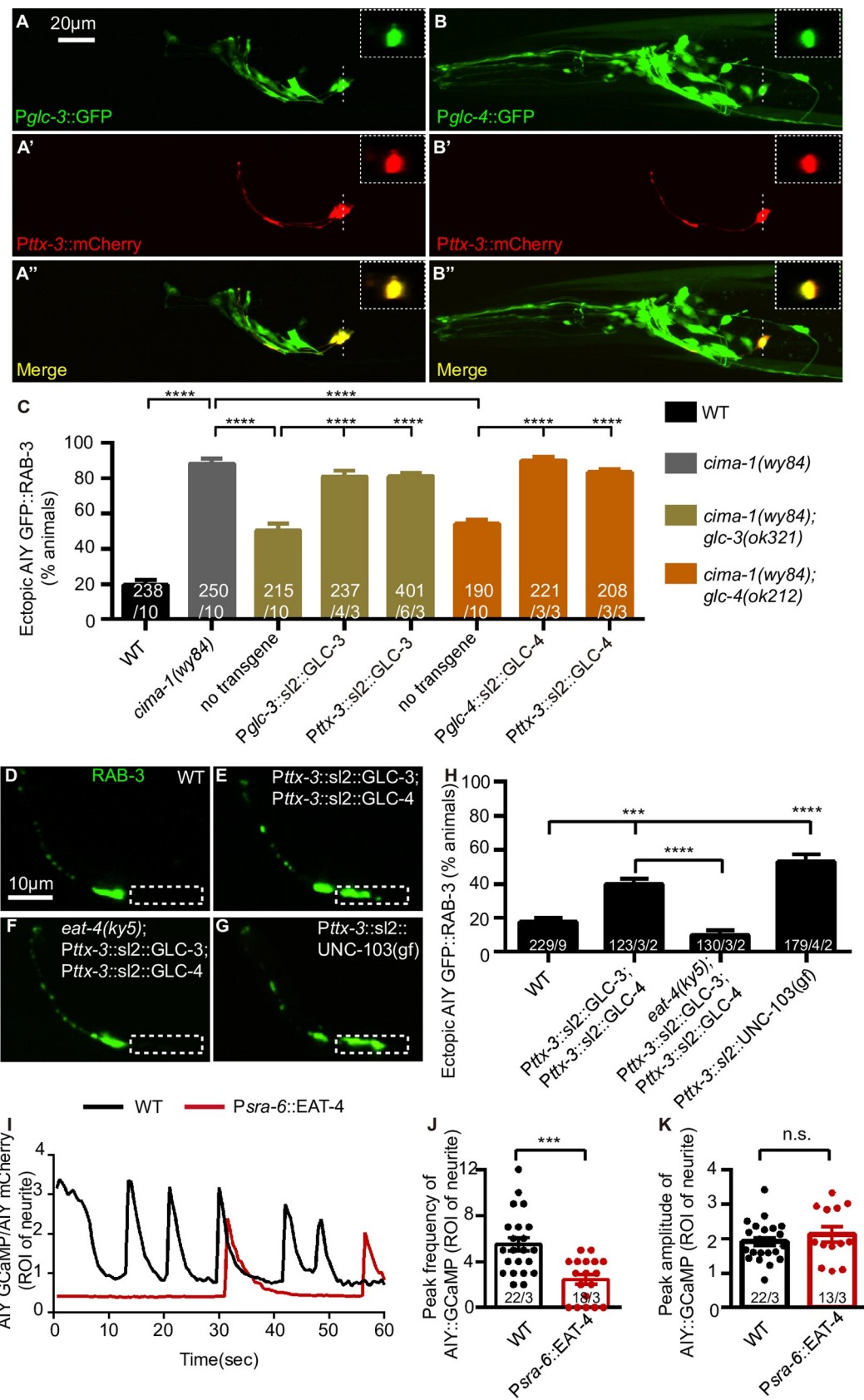

**Fig 4. GLC-3 and GLC-4 act cell-autonomously in the AIY to promote the ectopic synapse formation. (A-B")** Representative confocal micrographs of *glc-3* transcriptional reporter (P*glc-3*::GFP) (A), *glc-4* transcriptional reporter (P*glc-4*::GFP) (B) and AIY cytoplasmic marker (P*ttx-3*::mCherry) (A', B') at the adult Day 1 stage of wild-type worms. A" and B" are the merged graphs. The scale bar in (A) is 20μm and applies to (A'-A", B-B"). The dashed lines mark the position of the cross section of AIY cell body. The cross sections are displayed in the dashed boxes in the top-right of same panel. **(C)** Quantification of the percentage of animals displaying ectopic AIY presynaptic sites in the zone 1 region for indicated genotypes. The data show that AIY-specific expression of *glc-3* or *glc-4* rescues the corresponding mutation, indicating that GLC-3 and GLC-4 both act cell-autonomously in AIY. **(D-G)** Representative confocal micrographs of AIY presynaptic marker GFP::RAB-3 in wild-type animals (D), AIY-specific *glc-3* and *glc-4* overexpression in wild-type (E), *eat-4(ky5)* (F) and the AIY-specific *unc-103(gf)[UNC-103(A334T)]* animals (G). The dashed boxes indicate the zone 1 region. The scale bar in (D) is 10μm, applying to (E-G). **(H)** Quantification of the percentage of animals with ectopic AIY presynaptic sites corresponding to (D-G). The data suggest that overexpressing GLC-3 and GLC-4 simultaneously induces the ectopic synaptic formation, which requires *eat-4*. Moreover, inhibition of the AIY activity through expressing *unc-103(gf)* is sufficient to induce the ectopic synaptic formation. For (C) and (H), the total number of independent animals (N) and the number of biological replicates (n1) are indicated in each bar for each genotype. And for the transgenic lines created in (C) and P*ttx-3*::sl2::UNC-103(gf) in (G), the number of independent transgenic lines (n2) examined is indicated as the convention N/n1/n2. Statistics are based on one-way ANOVA with Dunnett's test. Error bars are SEM. \*\*\*P< 0.001, \*\*\*\*P< 0.0001. **(I)** The relatively AIY::GCaMP fluorescent signals of representative wild-type and P*sra-6*::EAT-4 transgenic animals over 60 seconds. The region of interesting (ROI) is circled by dashed line in S1 Video. Each data point is the ratio of AIY::GCaMP to AIY::mCherry. The frequency of $Ca^{2+}$ oscillation, but not the amplitude is dramatically reduced by the *eat-4(OE)*. **(J and K)** The GCaMP oscillation frequency (J) and amplitude (K) of relatively AIY::GCaMP fluorescent signals of wild-type and the ASH-specific *eat-4(OE)* (P*sra-6*::EAT-4) transgenic animals over 60 seconds. For J and K, each data point represents one independent animal. The total number of independent animals (N) and the number of biological replicates (n) are indicated in each bar for each genotype as N/n. Statistics are based on unpaired t test. Error bars are SEM. \*\*\*P< 0.001, n.s., not significant.

Together, these data show that ASH neurons are AIY presynaptic partners, which suggests that the formation of the ectopic AIY presynaptic structure may be due to the ectopic synaptic connections between ASH and AIY.

## High temperature alters synaptic subcellular specificity through glutamatergic signaling

To understand whether there is any physiological condition that can affect the AIY synaptic specificity, we tested the cultivation temperature since AIY is part of the thermotaxis circuit [46,51]. We examined the AIY presynaptic markers at a high physiological temperature (25°C) (Fig 7A). Wild-type animals can grow and reproduce normally at 25°C [96]. We found that the AIY morphology appeared largely intact at 25°C (S8A' and S8C' Fig). Interestingly, those animals displayed a highly penetrant ectopic synaptic structure as indicated by both GFP::RAB-3 and GFP::SYD-1 in the normally asynaptic zone 1 of AIY (GFP::RAB-3: 16.83 vs 79.67% at 22°C and 25°C respectively, p<0.0001; GFP::SYD-1: 15.79 vs 78.83% at 22°C and 25°C respectively, p<0.0001. Fig 7B–7F). Consistently, the ventral synaptic length and the ratio of the ventral to total synaptic length were increased at 25°C (8.53μm and 0.21 at 22°C; vs 16.88μm and 0.36 at 25°C, p<0.0001 for both comparisons. Fig 7G and 7H). The data indicate that high physiological temperature induces the ectopic synapses in AIY interneurons in wild-type animals.

Next, we asked whether the low temperature could inhibit the AIY ectopic synapses. To address this question, we quantified the AIY presynaptic phenotype in both wild-type and *cima-1(wy84)* animals at 15°C and 22°C. We found that the ectopic synapses were indeed reduced both in the wild-type and *cima-1(wy84)* animals at 15°C as compared to that at 22°C (S9A and S9B Fig). To exclude the possibility that the phenotypic difference was due to the slow development rate at 15°C, we also quantified the synaptic phenotype at the adult Day 2 stage and found similar results (S9A and S9B Fig). Those data indicate that high temperature promotes, while low temperature suppresses the AIY ectopic synaptic assembly.

To determine the temporal window required for high temperature to promote the AIY ectopic synapse formation, animals were shifted to 25°C during different developmental stages

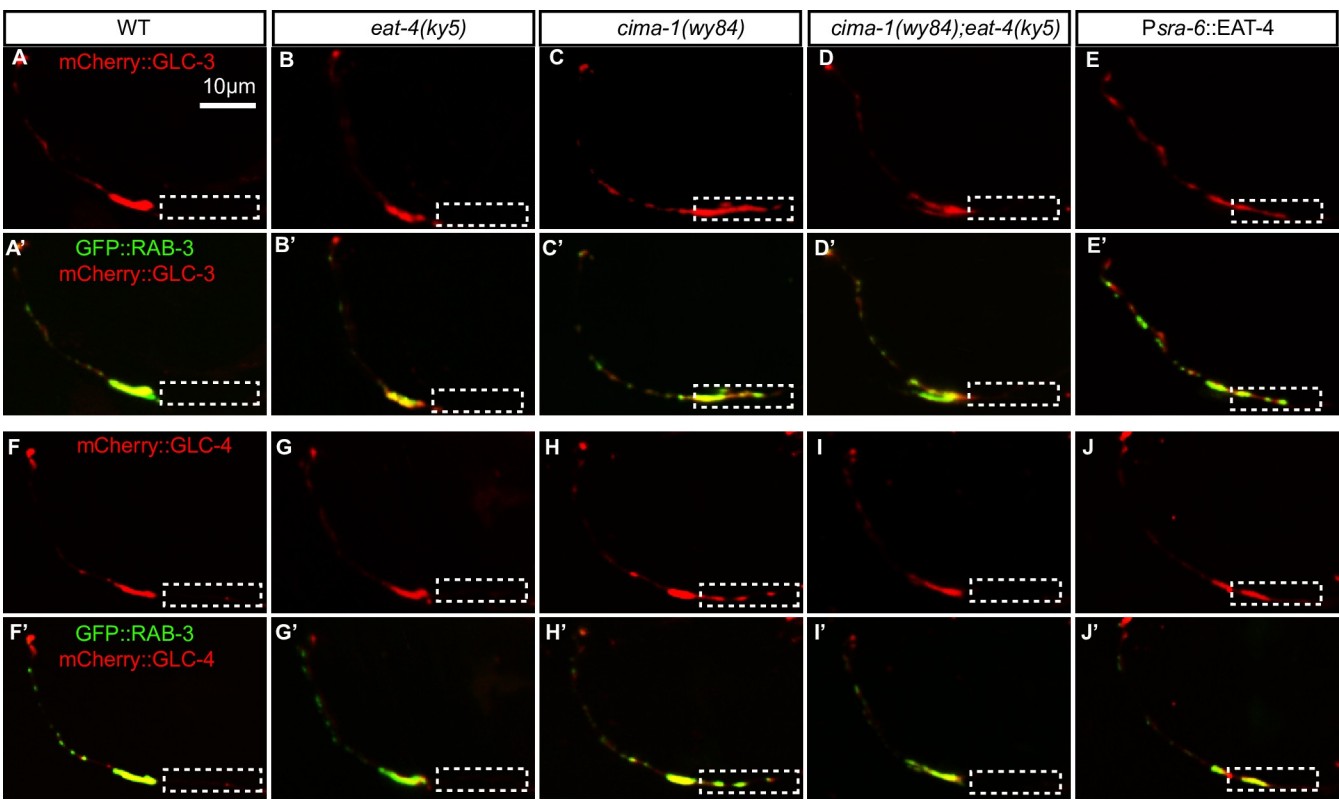

**Fig 5. GLC-3 and GLC-4 are enriched at the synaptic region in AIY interneurons. (A-E')** Representative confocal micrographs of mCherry::GLC-3 and GFP::RAB-3 double labeling in AIY interneurons. The mCherry::GLC-3 (A-E) and GFP::RAB-3 are partially colocalized in wild type (A'), *eat-4(ky5)* (B'), *cima-1(wy84)* (C'), *cima-1(wy84);eat-4(ky5)* (D') and the ASH-specific *eat-4* overexpressed animals (E'). **(F-J')** Representative confocal micrographs of mCherry::GLC-4 and GFP::RAB-3 double labeling in AIY interneurons. The mCherry::GLC-4 (F-J) and GFP::RAB-3 are partially colocalized in wild type (F'), *eat-4(ky5)* (G'), *cima-1(wy84)* (H'), *cima-1(wy84);eat-4(ky5)* (I') and the ASH-specific *eat-4* overexpressed animals (J'). GLC-3 and GLC-4 are ectopically localized to the zone 1 in *cima-1(wy84)* or ASH-specific *eat-4* overexpressing animals. Scale bar in (A) is 10μm and applies to all images in Fig 5.

(S9C Fig). Interestingly, the AIY ectopic synapse formation required the high temperature treatment during developmental stages, with more robust effect during embryonic stages (S9D Fig). No ectopic synapse was observed when treating from the larval L4 stage (S9D Fig). The results suggest that the AIY ectopic synaptic formation induced by high temperature is development-dependent.

Given that glutamate signaling is required for the AIY ectopic synaptic formation in *cima-1* mutants, we asked whether it was also required for the high temperature induced ectopic synapse formation. We examined the phenotype of *eat-4(ky5)* mutants at 25°C. Interestingly, *eat-4(ky5)* suppressed the AIY ectopic synapse formation at high temperature (82.43% and 20.81% of animals with ectopic synapses in wild-type and *eat-4(ky5)* mutants respectively, p<0.0001, Fig 7I, 7J and 7Q). The data demonstrate that glutamatergic neurotransmission is required for the AIY ectopic synaptic formation at high temperature.

Next, we determined whether GLC-3 and GLC-4 were required by examining the mutant phenotype at 25°C. Indeed, either *glc-3(ok321)* or *glc-4(ok212)* mutation partially, while the *glc-3(ok321);glc-4(ok212)* double mutations completely inhibited the ectopic synapses at 25°C (Fig 7K, 7L, 7M and 7Q). These results indicate that high temperature induces the AIY ectopic synaptic formation mediated by the glutamatergic GLC-3/GLC-4 receptors.

Given that the AIY ectopic synaptic formation in *eat-4(OE)* or *cima-1* mutant animals require glutamate transmission from ASH neurons, we asked whether ASH neurons were also

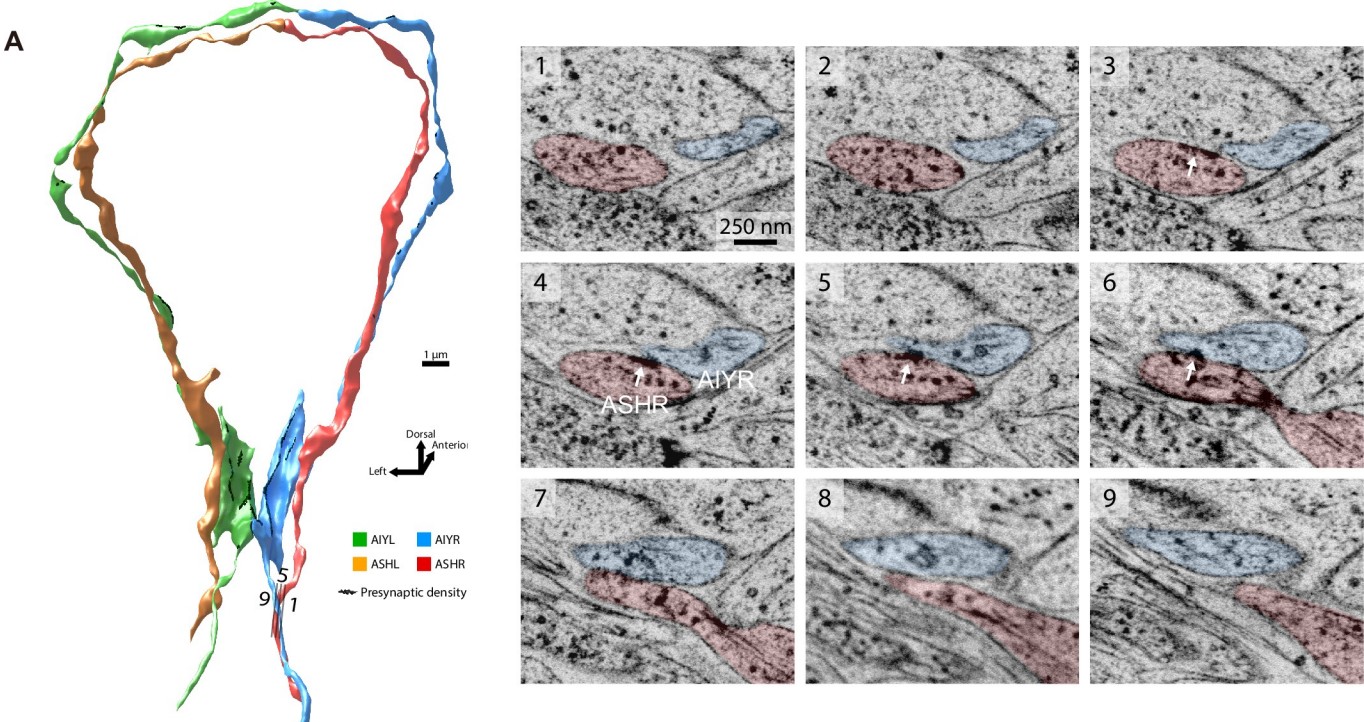

**Fig 6. ASH neurons are AIY presynaptic partners. (A)** Left: the 3D model shows the anatomic relationship between ASH and AIY. Right: nine consecutive high-resolution EM micrographs (slide 1, 5, and 9 are labeled in the 3D model) from an adult hermaphrodite show the synaptic connection between AIY and ASH at the anterior region of zone 1, near zone 2. Identified synapses from ASH to AIY are labeled with an arrowhead (image 3, 4, 5, 6). Scale bars are 1µm (left) and 250nm (right).

required for the high temperature induced AIY ectopic synaptic formation. Through cell specific *eat-4* rescue experiments, we found that expressing *eat-4* specifically in the ASH neurons significantly restored the ectopic synapses in *eat-4(ky5)* mutants at 25°C, which was more robust than that at 22°C (Fig 7N–7P and 7R), suggesting that ASH neurons are involved in high temperature induced AIY ectopic synaptic formation.

To visualize the anatomic relationship between ASH and AIY at high temperature, we labeled the ASH and AIY with cytoplasmic GFP and mCherry simultaneously, and found ASH process extended posteriorly alongside the AIY zone 1 (S8C-8C" Fig), suggesting that ASH could form synapses onto AIY in this region.

To address whether high temperature enhances the glutamate release from ASH, we quantified the intensity of the ASH VGLUT-pHluorin. PHluorin is a fluorescent protein quenched in acidic conditions such as inside the synaptic vesicle lumen [97]. We found that VGLUT-pHluorin intensity was enhanced in the ASH axon at high temperature, suggesting more glutamate vesicles releasing from ASH neurons (Fig 7S–7U). These results are consistent with the model that high temperature induces the AIY ectopic synaptic formation by enhancing the ASH glutamatergic neurotransmission.

Although 25°C is at the border line of the normal breeding temperature range (15–25°C), this could be a potential stress condition. To address if other stress conditions could induce the ectopic synapse formation, we tested the effect of osmotic and oxidative stresses on the AIY synaptic subcellular specificity, and found that animals treated with 200~500mM sorbitol or 0~10mM hydrogen peroxide displayed normal AIY synaptic distribution (S10 Fig), suggesting that the ectopic AIY synapses are not induced by general stresses.

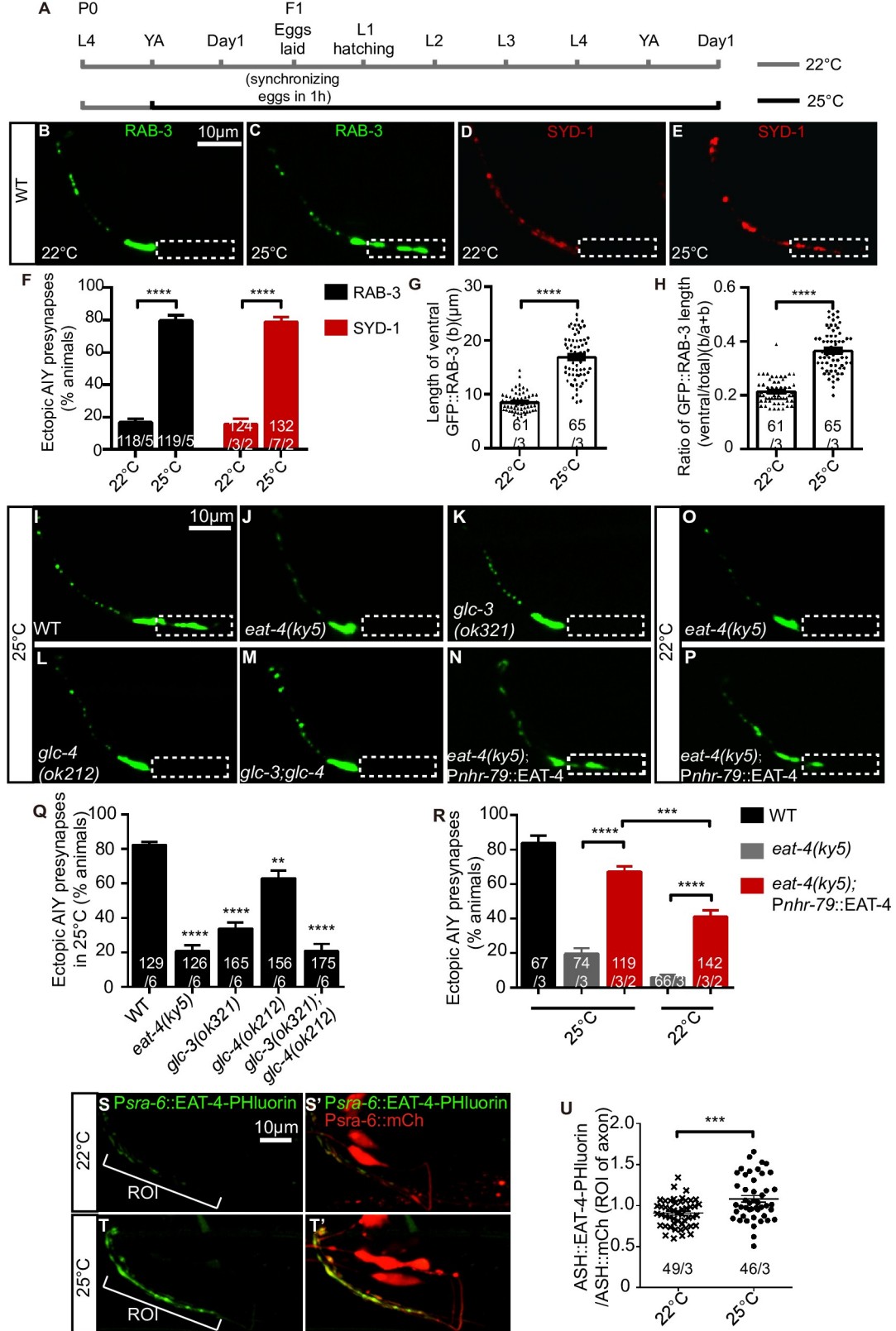

**Fig 7. High temperature disrupts the synaptic subcellular specificity mediated by EAT-4, GLC-3 and GLC-4. (A)** A schematic diagram shows the cultivation temperature conditions. The control group were cultivated at the constant 22˚C (gray

line). The high temperature group was transferred from 22˚C (gray line) into 25˚C (black line) since the parent generation (P0) young adult stage (YA: 12 hours post larval stage 4) until the next generation (F1) adult Day 1 stage when the phenotype was scored. **(B-E)** Representative confocal micrographs of the AIY synaptic vesicle marker GFP::RAB-3 (B and C) or active zone marker GFP::SYD-1 (pseudo-red, D and E). When cultivated at 22˚C, the AIY presynaptic distribution is normal, as indicated with GFP::RAB-3 (B) and GFP::SYD-1 (D). However, when cultivated at 25˚C, the ectopic synapses emerge in the zone 1 region (C, E). Dashed boxes indicate the zone 1 region of AIY. The scale bar in (B) is 10μm and applies to (C-E). **(F-H)** Quantification of the percentage of animals with ectopic AIY synaptic vesicle GFP::RAB-3 (black bars) and active zone GFP::SYD-1 (red bars) (F), the ventral synaptic length (G) and the ratio of the ventral to the total synaptic length (H). Both (G) and (H) are based on the GFP::RAB-3, and each spot represents the value from one independent AIY. The total number of independent animals (N) and the number of biological replicates (n1) are indicated in each bar for each genotype as N/n1. And for the transgenic lines created in F, the number of independent transgenic lines (n2) examined is indicated as the convention N/n1/n2. Error bars are SEM. \*\*\*\*P< 0.0001. Statistics are based on unpaired t test. **(I-P)** Representative confocal micrographs of the AIY GFP::RAB-3 in wild-type (I), *eat-4(ky5)* (J, O), *glc-3(ok321)* (K), *glc-4(ok212)* (L), and *glc-3(ok321);glc-4(ok212)* (M), *eat-4(ky5)* with ASH-specific expressing *eat-4* (P*nhr-79*) transgenes (N, P) at 25˚C (I, J, K, L, M) or 22˚C (O, P). Dashed boxes mark the zone 1 of AIY interneurons. The scale bar in (I) is 10μm and applies to (J-P). **(Q)** Quantification of the percentage of animals with ectopic AIY synaptic sites in the zone 1 region corresponding to (I-M). The data indicate that *eat-4 (ky5)*, *glc-3(ok321)* or *glc-4(ok212)* mutations robustly inhibit the ectopic synapse formation induced by high temperature (25˚C). **(R)** Quantification of the percentage of animals with ectopic synapses in AIY zone 1 region for the indicated conditions/genotypes. *eat-4* expressed in the ASH significantly restores the ectopic synapses in *eat-4(ky5)* mutants at 25˚C, which is more robust than that at 22˚C. For Q and R, the total number of independent animals (N) and the number of biological replicates (n1) are indicated in each bar for each genotype as N/n1. And for the transgenic lines created in R, the number of transgenic lines (n2) examined is indicated as the convention N/n1/n2. Error bars are SEM. \*\*P< 0.01, \*\*\*\*P< 0.0001. Statistics are based on one-way ANOVA with Dunnett's test. **(S-T')** Representative confocal micrographs of P*sra-6*::EAT-4-PHluorin and P*sra-6*::mCherry double labeling in wild-type animals cultivated at 22˚C(S, S') and 25˚C(T, T'). The ROI is the axon of ASH neurons which is marked by skewed bracket (S, T). The scale bar in (S) is 10μm and applies to (S', T-T'). **(U)** The relative ASH::EAT-4-PHluorin fluorescent intensity in wild-type animals cultivated in 22˚C and 25˚C. Each data point represents a single independent animal. The total number of independent animals (N) and the number of biological replicates (n) are indicated in each bar for each genotype as N/n. Error bars are SEM. \*\*\*P = 0.0002. Statistics are based on unpaired t test.

## Discussion

Our previous study identified that *cima-1* in epidermis is required for the normal AIY presynaptic distribution. *cima-1* functions partially through the VCSC glia [55]. In this study, we uncover an inhibitory glutamate signaling that is required for the *cima-1(wy84)*-induced AIY

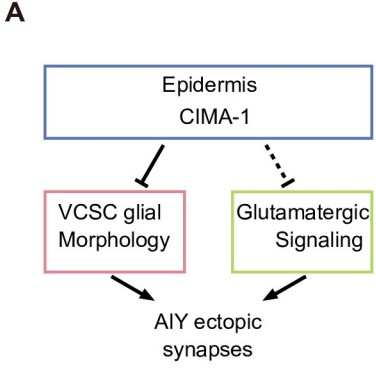

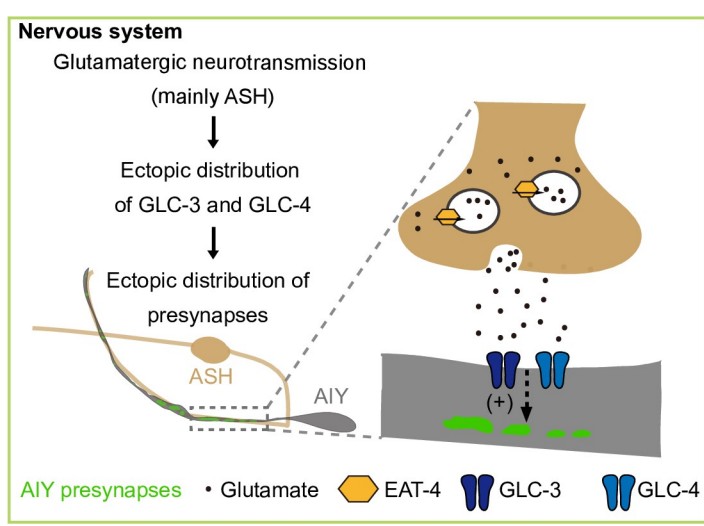

**Fig 8. A model explaining the AIY synaptic subcellular specificity. (A)** A model explaining the AIY synaptic subcellular specificity. CIMA-1 in epidermal cells regulates the AIY presynaptic subcellular specificity by two pathways: VCSC glia signaling and glutamatergic signals. The glutamatergic signaling, which can also be increased by *eat-4(OE)* or high cultivation temperature, promotes the ectopic distribution of GLC-3 (dark blue) and GLC-4 (light blue) receptors in the AIY zone 1 region, where these receptors regulate the ectopic presynaptic formation.

ectopic synaptic formation (Fig 8A). Furthermore, we show that *eat-4(OE)* or high temperature can trigger the glutamate signaling from ASH sensory neurons to promote the ectopic presynaptic formation, which is mediated by the inhibitory glutamate gated chloride channels GLC-3 and GLC-4 in the AIY interneurons. These findings describe a novel mechanism underlying synaptic subcellular specificity.

### ASH neurons form inhibitory synapses onto the AIY

In this study, we demonstrate that ASH forms inhibitory synapses on the AIY interneurons. Four lines of evidence support this. First, ASH processes are aligned next to the AIY, which indicates ASH may form synapses onto AIY (S8 Fig). Secondly, through tissue-specific expression analysis, we showed that the glutamate required for the AIY ectopic synaptic formation is released from the ASH neurons and sensed by the GLC-3/GLC-4 receptors in the AIY. Thirdly, ASH specific *eat-4(OE)* reduces the frequency of AIY $Ca^{2+}$ oscillation, indicating that ASH inhibits AIY excitability. Finally, the ASH-AIY synaptic connection was confirmed by electron microscopy reconstruction [95].

The next question is why expressing *eat-4* in other AIY presynaptic glutamatergic neurons such as AFD and AWC does not rescue. There are two possibilities. First, the amount or frequency of the glutamate released from ASH could be much higher than from any of other AIY presynaptic neurons. Second, the ASH-AIY synapses, which localizes at the border of zone 2 and zone 1 in the wild-type animals, are closer to the ectopic synaptic sites in the zone 1 than those of AFD-AIY or AWC-AIY. Therefore, the glutamate from ASH can diffuse more easily to the zone 1 region where it probably locates the GLC-3/GLC-4 receptors and promotes the ectopic synaptic assembly.

In vertebrates, excitatory neuronal activity is well recognized for its role in modulating excitatory synapse formation, maturation and plasticity [98–100]. More recently, GABAergic activity was also found to regulate both inhibitory and excitatory synaptic development at early developmental stage through depolarizing the postsynaptic neurons [101,102]. However, our knowledge about the role of GABA activity in promoting synaptic formation is largely limited to the early developmental stage when GABA acts as an excitatory transmitter [103]. In this study, we demonstrated that an important role of the inhibitory ASH-AIY synaptic transmission in promoting ectopic excitatory presynaptic assembly in the postsynaptic AIY neurons. The future work should focus on understanding the underlying molecular mechanisms.

### Pentameric ligand-gated ion channels regulate synaptic specificity

Glutamate signals promote the AIY ectopic synaptic formation through two pentameric ligand-gated ion channels GLC-3 and GLC-4, which are localized to the AIY presynaptic region, partially overlapping with the presynaptic marker RAB-3. Unlike a typical bipolar neuron, which assembles presynaptic and postsynaptic structures in axons or dendrites, AIY presynaptic and postsynaptic sites are overlapping along the single neurite in zone 2 and 3 regions [5]. The close anatomic relationship between postsynaptic and presynaptic sites may be helpful for the activity-dependent presynaptic assembly. Alternatively, GLC-3 and GLC-4 may also localize to the presynaptic sites. In this case, GLC-3 and GLC-4 may be activated by the glutamate spillover from adjacent synapses.

Glutamate spillover plays physiological or pathological roles [104–107]. The loss of astrocyte-like VCSC glia or glutamate reuptake transporter GLT-1 can alter the animal escaping or exploration behavior [107]. Increasing the extracellular level of glutamate may also result in neurotoxicity and degeneration [104,108]. Similar functions of glutamate present in mammals [105,106].

The inhibitory neurotransmitter receptors such as GABA receptors were also found in the excitatory presynaptic boutons in mammalian brain, where they play important roles in regulating synaptic transmission [109–112]. However, it is largely unknown if these presynaptic inhibitory receptors are involved in synaptic development or plasticity.

The closest related mammalian homologs of GLC-3 and GLC-4 are glycine receptors (GlyRs) [113–115]. GlyRs are one of the major inhibitory neurotransmitter receptors, involved not only in neuronal signaling processing, but also in neurodevelopment [116]. GlyRs regulate postsynaptic protein clustering in immature rat spinal neurons [117], and cortical interneuron migration in mouse [116]. Mutations of GlyRs are associated with a number of neurological disorders including hyperekplexia, temporal lobe epilepsy, chronic inflammatory pain, autism, etc, which makes GlyRs potential drug targets [118]. Given the functional conservation of pLGIC family receptors, *C. elegans* GLC-3 and GLC-4 may provide an excellent model to address the mechanisms underlying physiological and pathological roles of GlyRs.

## Temporal regulation of spatial specificity

During embryonic development, the AIY presynaptic assembly in zone 2 region is mainly regulated by netrin/DCC secreted from the VCSC glia [6]. However, it is largely unknown how the zone 1 avoids synaptic assembly. In this study, we found that the amount of glutamate released from ASH is critical for the synaptic assembly in zone 1 region. Although glutamatergic neurotransmission from ASH is also required for the ectopic synapse formation in *cima-1 (wy84)* mutants, we noticed that the synaptic subcellular defects are different between *cima-1 (wy84)* and *eat-4(OE)* animals. The ectopic synapses appear since newly hatched larval L1 stage in *eat-4(OE)* animals and at the adult stage in *cima-1* mutants [55]. Additionally, the VCSC glia contribute more to the synaptic defect of *cima-1(wy84)* than that of *eat-4(OE)*. Those differences indicate that *cima-1(wy84)* and *eat-4(OE)* may regulate the synaptic subcellular specificity through different molecular mechanisms.

## Environmental temperature affects synaptic subcellular specificity

In this study, we showed that the synaptic subcellular specificity was affected by temperature during developmental stages. Specifically, we showed that high temperature promoted the ectopic synaptic formation mediated by the vesicle glutamate transporter VGLUT/EAT-4 in ASH and glutamate receptors GLC-3/GLC-4 in AIY, while low temperature inhibited the ectopic synaptic assembly. This finding suggests that temperature modulates the synaptic subcellular specificity through glutamatergic neurotransmission. No ectopic synapse observed under osmotic or oxidative stresses suggests the synaptic specificity is not affected by general stresses.

The AIY interneurons are part of the thermosensory circuit involved in the thermotaxis behavior [46–51]. Previous studies have identified that AFD, AWC and ASI are major thermosensory neurons [46,51–54,119]. In this study, we found that ASH sensory neurons could sense the cultivation temperature and regulate the AIY synaptic subcellular specificity, suggesting that the ASH could be part of the thermosensory circuit, which should be further tested in the future.

Temperature is a common and vital environmental factor for many organisms. The nervous system is very sensitive to high temperature during embryogenesis [120]. High temperature often results in neurological disorders including neural tube defects, microcephaly, microphthalmia, microvascular abnormity in vertebrates [120]. In *Drosophila*, high temperature also induces neural developmental defects [21,23,121,122]. Temperature can modulate the nematode *C. elegans* thermotaxis behaviors and lifespan mediated by neuronal activity [123–125]. In our study, the effects of temperature on synaptic subcellular specificity provide an excellent

model to address the mechanistic insights into the high temperature induced neurodevelopmental defects in *vivo*.

## Materials and methods

### Strains and cultivation

Strains were cultivated on OP50-seeded nematode growth medium (NGM) plates at 22°C unless specified [126]. Wild-type (WT) animals are Bristol strain N2. Mutant alleles used in this study include:

LGI: *unc-13(e1091), avr-14(ad1302), glc-2(gk179), mgl-2(tm355), glr-3(tm6403)*
LGII: *cat-2(e1112), glc-4(ok212), nmr-1(ak4), glr-4(tm3239)*
LGIII: *eat-4(ky5), eat-4(nj2), eat-4(nj6), unc-47(n2409), glr-1(n2461), glr-2(ok2342)*
LGIV: *cima-1(wy84), cima-1(gk902655), unc-17(cn355), mgl-3(tm1766)*
LGV: *avr-15(ad1051), glc-1(pk54), glc-3(ok321), nmr-2(ok3324), glr-5(tm3506)*
LGX: *mgl-1(tm1811), glr-6(tm2729), glr-7(tm1824)*
All worm strains used in this study are listed in the S1 Excel.

### Plasmids and transgenic manipulations

Plasmids were made in the pSM or pPD49.26 by recombination [127]. The transgenic strains carrying extrachromosomal DNA arrays were generated using standard microinjection protocol [128]. The following plasmids were used as co-injection markers: P*hlh-17*::mCherry, P*myo-3*::mCherry, P*unc-122*::GFP, P*unc-122*::RFP or P*lin44*::mCherry. Unless otherwise stated in S1 Excel, the concentration of plasmids was injected at 20 ng/μl. The cDNA plasmids generated for use in this study (*glc-3* cDNA, *glc-4* cDNA), were cloned by RT-PCR from total RNA isolated from WT (N2) worms. The *unc-103*$^{A334T}$ cDNA was amplified from the strain SQC0132 [*yfhIx0132* (P*unc-103*::*unc-103*$^{A334T}$::GFP)] [92], which is a gift from Dr. Shiqing Cai. The *eat-4* cDNA was cloned from the plasmid P*eat-4a*::*eat-4* (cDNA)::GFP [51] from Dr. Ikue Mori. The P*sra-6*::caspase p12 and P*sra-6*::caspase p17 constructs were modified from plasmids DACR336(P*ttx-3*::caspase p12) and DACR335(P*ttx-3*::caspase p17) respectively through replacing the *ttx-3* promoter with *sra-6* promoter (4kb) by recombination[50,83].

The ASH-specific EAT-4(VGLUT)-pHluorin expression construct was created through inserting the PHluorin CDS into the P*sra-6*::*eat-4a*[129]. The PHluorin was inserted after the conserved glycine residue at position 106 of *eat-4* A isoform cDNA by PCR primers which adds 42 bases (TCTACCTCTGGAGGATCTGGAGGAACCGGAGGATCTATGGGA) for the upstream linker, 45 bases (ACCGGTGGAGGAACCGGAGGAACCGGAGGA TCTGGAGG AACCGGA) for downstream linker, as previously described [129]. Forward primer to amplify PHluorin: 5'- GAGGATCTGGAGGAACCGGAGGATCTATGGGAAGTAAAGGAGAAG AACTTTTC-3'. Reverse primer to amplify PHluorin: 5'- CTCCAGATCCTCCGGTTCCTCC GGTTCCTCCACCGGTTTTGTATAGTTCATCC-3'. The vector were amplified from the P*sra-6*::*eat-4* plasmids with forward primer: 5'-ACCGGAGGAACCGGAGGATCTGGAGGAA CCGGAAAAGTTCAT ATGCATGAATTC-3' and reverse primer: 5'-GATCCTCCGGTTCC TCCAGATC CTCCAGAGGTAGATCCGTATGGATCTGTATAATTTT-3'. All plasmids and primer information in this study were listed in the S2 Excel.

### ASH and glia ablation

The two-component system of reconstituted caspase (recCaspase) [83] was driven by the *sra-6* promoter, which specifically ablating the ASH neurons. Ablation was confirmed by lack of ASH specific marker (*kyIs39*) [78].

The two-component system of reconstituted caspase (recCaspase) [83] was driven by the *hlh-17* promoter, which specifically ablating the CEPsh glia. Ablation was confirmed by lack of the CEPsh-specific marker(*nsIs105*) [107].

### Electron microscopy analysis

Serial-section electron microscopy datasets were imported into CATMAID [130] to peruse. Each section containing AIY was examined to determine if contact was made with ASH, and if so, whether chemical synapses were present. Chemical synapses were defined as a presynaptic bouton containing a pool of synaptic vesicles as well as a dense presynaptic projection inside the membrane.

### Special temperature treatment

Animals was transferred to 15°C or 25°C at specific time points as illustrated in the figures. The phenotype of next or the same generation was scored at the adult Day 1 or Day 2 stage. In these assays, animal synchronization was done through two steps. First, eggs were collected within one-hour time window; second, animals were synchronized at the L4 stage.

### Osmotic stress assays

Young adults were grown on NGM agar plates containing 0mM(control), 200mM, 300mM, 400mM or 500 mM sorbitol seeded with OP50 until they reached the adult Day 1 stage when the synaptic phenotype was scored. The concentrations and methods were modified from the study of Chandler-Brown et al. [131].

### Oxidative stress assays

Young adults were grown on NGM agar plates with OP50 and supplemented with S-basal buffer containing hydrogen peroxide (0.5mM, 2mM, 5mM,10 mM) at specific time points as illustrated in the figures. Animals were synchronized at the L4 stage and phenotypes were scored 24 hours later. The concentrations were modified from Lee *et al.* [132]. Animals can survive and reproduce at low concentration of $H_2O_2$ (0.5mM, 2mM) from the young P0 stage, but not at higher than 5mM. We also treated animals with high concentrations (5mM, 10mM) for shorter time (time window4 in the high temperature treatment).

### Calcium imaging of AIY neurons

For *in vivo* calcium imaging, individual Day 1 (D1) adult hermaphroditic worms were immobilized with Polybead Microspheres 0.10μm (Polysciences) on 12% agarose pads. Fluorescent images were acquired using an Andor Dragonfly Spinning Disc Confocal Microscope with 60x objectives coupled with an ZYLA camera. GCaMP6s (in AIY) was excited by 488nm excitation wavelength lasers, and the mCherry control was imaged with 561 nm excitation wavelength lasers. The fluorescent signals of video were collected at the rate of 2 Hz[133].

For AIY GCaMP signals, the ROI is AIY neurite (Zone 2 and Zone 1). The relatively GCaMP signals for each data point were calculated as:

$$F_{GCaMP} / F_{mCherry}$$

$$F_{GCaMP} = \text{average GCaMP fluorescence of the ROI at a time point}$$

$$F_{mCherry} = \text{average mCherry fluorescence of the ROI at a time point}$$

For peak frequency of AIY GCaMP was taken as $F_n$, which was calculated as:

$$F_n = \text{scintillation times of AIY GCaMP in 1 minute}$$

For peak amplitude of AIY GCaMP was calculated as:

$$((F_{max(1)} - F_{min}) + \cdots + (F_{max(n)} - F_{min})) / F_n$$

$$F_{max(1)} = \text{the highest relatively GCaMP signals of the ROI at first scintillation}$$

$$F_{max(n)} = \text{the highest relatively GCaMP signals of the ROI at n scintillation}$$

$$F_{min} = \text{the lowest relatively GCaMP signals of the ROI in 1 minute}$$

The data of fluorescence intensity was quantified with the ImageJ (Fiji).

## Fluorescence microscopy and confocal imaging

Confocal images were acquired with an Andor Dragonfly Spinning Disc Confocal Microscope with 40x or 60x objectives. The fluorescently tagged fusion proteins GFP or mCherry was imaged with 488 or 561 nm excitation wavelength lasers, respectively. Animals were anesthetized with 50mM muscimol or Polybead Microspheres 0.10μm (the recorded about GCaMP and PHluorin). Images were processed with Imaris, ImageJ (Fiji) and Photoshop. All images are oriented anterior to the left and dorsal up.

## Quantification and statistical analysis

To quantify the percentage of animals with ectopic synapses of AIY zone 1 at the adult stage, animals were synchronized at larva stage 4 (L4) and then we scored the phenotypes 24 hours later using a Nikon Ni-U fluorescent microscope with 40x objectives or Andor Dragonfly Spinning Disc Confocal Microscope with 40x objectives. For the larval phenotypes, synchronized eggs were cultivated for 12 and 48 hours to reach the middle stage of L1 and L4. At least three biological replicates were done for each quantification. For transgenic analysis, at least two independent transgenic lines were generated and quantified unless specified. The data of AIY ectopic synapses were blindly recorded. Other data were collected based on genotypes or treatments. All quantitative raw data are in S3 Excel.

For ASH EAT-4-PHluorin intensity, the ROI is ventral axon of ASH. The relatively PHluorin intensity for each data point were calculated as: $F_{PHluorin}/F_{mCherry}$.

$$F_{PHluorin} = \text{average PHluorin fluorescence of the ROI}$$

$$F_{mCherry} = \text{average mCherry fluorescence of the ROI}$$

The data of fluorescence intensity was collected using the ImageJ (Fiji).

Statistical analyses were conducted with GraphPad Prism software (version 6.01). The comparisons between two groups were determined by the unpaired t test, while multiple comparisons were analyzed with one-way analysis of variance with Dunnett's multiple comparison test. Error bars represent the standard errors of the mean (SEM).

## Supporting information

**S1 Fig. Neurotransmission are not required for synaptic subcellular specificity per se. (A)**
Diagrams of the *unc-47*, *unc-17* and *cat-2* genomic structures, respectively. Exons and introns
are indicated by boxes (yellow boxes are translated regions; gray boxes are untranslated
regions) and black lines. Mutations are marked with asterisks. **(B-I)** Representative confocal
micrographs of the AIY synaptic GFP::RAB-3 in wild-type (A), *unc-13(e1091)* (B), *eat-4(ky5)*
(C), *eat-4(nj2)* (D), *eat-4(nj6)* (E), *unc-47(n2409)* (F), *unc-17(cn355)* (G) and *cat-2(e1112)* (H)
animals at the adult Day 1 stage. Dashed boxes mark the zone 1 of AIY interneurons. The scale
bar in (A) is 10μm and applies to (B-H). **(J)** Quantification of the percentage of animals with
ectopic AIY synaptic marker GFP::RAB-3 in the zone 1 region for the indicated genotypes.
The total number of independent animals (N) and the number of biological replicates (n) are
indicated in each bar for each genotype (N/n). Statistics are based on one-way ANOVA with
Dunnett's test. Error bars are SEM. n.s., not significant.
(TIF)

**S2 Fig. GABAergic, cholinergic and dopaminergic neurotransmissions are not required
for the ectopic synaptic formation in *cima-1(wy84)*. (A-D)** Representative confocal micro-
graphs of the AIY synaptic marker GFP::RAB-3 in *cima-1(wy84)* (B), *cima-1(wy84);unc-47
(n2409)* (C), *cima-1(wy84);unc-17(cn355)* (D) and *cima-1(wy84);cat-2(e1112)* mutant (E) adult
Day 1 animals. Dashed boxes mark the zone 1 region of AIY interneurons. The scale bar in (B)
is 10μm and applies to (C-E). **(E)** Quantification of the percentage of animals with ectopic AIY
synaptic marker GFP::RAB-3 in the zone 1 region. Note the ectopic synapses in *cima-1(wy84)*
are not suppressed by mutations disrupting GABAergic (*unc-47(n2409)*), cholinergic (*unc-17
(cn355)*) or dopaminergic (*cat-2(e1112)*) synaptic transmission. In the graph, the total number
of independent animals (N) and the number of biological replicates (n) are indicated in each
bar for each genotype as N/n. Statistics are based on one-way ANOVA with Dunnett's test.
Error bars are SEM. n.s., not significant.
(TIF)

**S3 Fig. The expression of *eat-4, glc-3, glc-4* begins at the embryo stage. (A-B')** A representa-
tive confocal micrograph of *eat-4* translational reporter (P*eat-4a*::*eat-4*::GFP). The expression
of the reporter is enriched in the nervous system at the adult stage (A) and embryonic stage
(B'). (B) is the corresponding bright field micrograph. **(C and C')** A representative confocal
micrograph of *glc-3* transcriptional reporter (P*glc-3*::GFP) at the embryonic stage (C') and the
corresponding bright field micrograph (C). **(D and D')** A representative confocal micrograph
of *glc-4* transcriptional reporter (P*glc-4*::GFP) at the embryonic stage(D') and the correspond-
ing bright field micrograph (D). The scale bars are 10μm, and the one in (B) applies to (B', C,
C', D, D').
(TIF)

**S4 Fig. The AIY ectopic synapses induced by *eat-4*(OE) is largely independent of the VCSC
glia. (A)** Quantification of the percentage of animals with the ectopic AIY synaptic GFP::RAB-
3 in the zone 1 region for the indicated genotypes. The data showed that VCSC glia only con-
tribute partially to the synaptic subcellular specificity defect in either *cima-1(wy84)* or *eat-4
(OE)* (P*eat-4a*::EAT-4) strains. Error bars are SEM. \*\*P< 0.01, \*\*\*\*P< 0.0001, n.s., not signifi-
cant. Statistics are based on one-way ANOVA with Dunnett's test (the group of glia ablation)
or unpaired t test (between the control group and the corresponding group of glia ablation).
The total number of independent animals (N) and the number of biological replicates (n1) are
indicated in each bar for each genotype, as are, for the transgenic lines created, the number of

independent transgenic lines (n2) examined (using the convention N/n1 or N/n1/n2).
(TIF)

**S5 Fig.** *eat-4(OE)* **promotes the AIY ectopic synapse formation since L1 stage. (A-F)** Representative confocal micrographs of the AIY presynaptic marker GFP::RAB-3 in *eat-4(OE)* animals at different developmental stages. The presynaptic marker is not present in zone 1 region at larval L1 (A), L4 (C) or adult Day 1 stages (E) in wild type. However, the ectopic synapses appear in *eat-4(OE)* animals at larval L1 (B), L4 (D) and adult Day 1 stages (F), as indicated in the dashed boxes. Dashed boxes mark the zone 1 of AIY interneurons. The scale bars are 10μm, and the one in (A) applies to (B), in (C) applies to (D-F). **(G-I)** Quantification of the percentage of animals with the ectopic synapses in the AIY zone 1 (G), the ventral presynaptic length (H), and the ratio of the ventral to total presynaptic length (I) based on GFP::RAB-3. All quantification data consistently indicate that *eat-4(OE)* induces ectopic synapses since the newly hatched larval L1 stage. For (H) and (I), each spot represents the value from a single AIY. In the graph, the total number of independent AIY or animals (N) and the number of biological replicates (n1) are indicated in each bar for each genotype as N/n1. And for the transgenic lines created, the number of independent transgenic lines (n2) examined indicated in each bar for each genotype as N/n1/n2. For (H) and (I), one of transgenic lines in (G) was measured. Statistics are based on unpaired t test. Error bars are SEM. ****P< 0.0001. **(J-L)** Simultaneous visualization of GFP::RAB-3 in AIY and the postsynaptic RIA neurons (P*glr-3*::mCherry) in wild-type animals cultivated at 22˚C (J), 25˚C (L) and *eat-4(OE)* animals (K). The arrows indicate the posterior endpoint of RIA. The AIY presynapses extend beyond the RIA endpoint in wild-type animals cultivated at 25˚C (L) and *eat-4(OE)* animals (K).
(TIF)

**S6 Fig. Glutamate receptors are not required for AIY synaptic subcellular specificity per se.** (**A-Q**) Representative confocal micrographs of AIY presynaptic marker GFP::RAB-3 in wild-type (A), *glr-1(n2461)* (B), *glr-2(ok2342)* (C), *glr-3(tm6403)* (D), *glr-4(tm3239)* (E), *glr-5 (tm3506)* (F), *glr-6(tm2729)* (G), *glr-7(tm1824)* (H), *nmr-1(ak4)* (I), *nmr-2(ok3324)* (J), *mgl-1 (tm1811)* (K), *mgl-2(tm355)* (L), *mgl-3(tm1766)* (M), *avr-14(ad1302);avr-15(ad1501);glc-1 (pk54)* (N), *glc-2(gk179)* (O), *glc-3(ok321)* (P), *glc-4(ok212)* (Q) animals. In all images, dashed boxes correspond to zone 1 of AIY interneurons. The scale bar in (A) is 10μm, applying to (B-Q). **(R)** Quantification of the percentage of animals with the ectopic AIY synaptic marker GFP::RAB-3 corresponding to (A-Q). The data show that none of those glutamate receptors is required for synaptic subcellular specificity per se. The total number of independent animals (N) and the number of biological replicates (n) are indicated in each bar for each genotype (N/n). Statistics are based on one-way ANOVA with Dunnett's test. Error bars are SEM. n.s., not significant.
(TIF)

**S7 Fig. Glutamate-gated chloride channels GLC-3 and GLC-4 are required for the ectopic synapse formation in *cima-1(wy84)*. (A-T)** Representative confocal micrographs of AIY presynaptic marker GFP::RAB-3 in wild type (A), *cima-1(wy84)* (B), *cima-1(wy84); glr-1(n2461)* (C), *cima-1(wy84);glr-2(ok2342)* (D), *cima-1(wy84);glr-3(tm6403)* (E), *cima-1(wy84); glr-4 (tm3239)* (F), *cima-1(wy84); glr-5(tm3506)* (G), *cima-1(wy84); glr-6(tm2729)* (H), *cima-1 (wy84); glr-7(tm1824)* (I), *cima-1(wy84); nmr-1(ak4)* (J), *cima-1(wy84);nmr-2(ok3324)* (K), *cima-1(wy84);mgl-1(tm1811)* (L), *cima-1(wy84);mgl-2(tm355)* (M), *cima-1(wy84);mgl-3 (tm1766)* (N),*cima-1(wy84); avr-14(ad1302)* (O), *cima-1(wy84);avr-15(ad1501)* (P), *cima-1 (wy84);glc-1(pk54)* (Q), *cima-1(wy84);glc-2(gk179)* (R), *cima-1(wy84);glc-3(ok321)* (S), *cima-1 (wy84);glc-4(ok212)* (T). GLC-3 and GLC-4 partially mediate the ectopic presynaptic specificity

in *cima-1(wy84)*. In all images, dashed boxes correspond to zone 1 of AIY interneurons. The scale bar in (A) is 10μm, applying to (B-T). **(U)** Quantification of the percentage of animals with ectopic AIY synaptic marker GFP::RAB-3 in the zone 1 region corresponding to (A-T). **(V-Y)** Representative confocal micrographs of AIY presynaptic marker GFP::RAB-3 in *cima-1 (gk902655)* (V), *cima-1(wy84);glc-3(ok321)* (W), *cima-1(wy84);glc-4(ok212)* (X), *cima-1(wy84); glc-3(ok321);glc-4(ok212)* (Y) mutants. The dashed boxes correspond to zone 1 of AIY interneurons. The scale bar in (V) is 10μm, applying to (W-Y). **(Z)** Quantification of the percentage of animals with the ectopic AIY synaptic marker GFP::RAB-3 in the zone 1 region corresponding to (V-Y). For U and Z, the total number of independent animals (N) and the number of biological replicates (n) are indicated in each bar for each genotype (N/n). Statistics were based on one-way ANOVA with Dunnett's test. Error bars are SEM. $^{**}P < 0.01$, $^{***}P < 0.001$, $^{****}P < 0.0001$, n.s., not significant.
(TIF)

**S8 Fig. The ASH axons are extended posteriorly and overlap with AIY zone 1 in *cima-1 (wy84)* and 25˚C treated wild-type animals. (A-C")** Representative confocal micrographs of ASH (P*nhr-79*::GFP) (A, B, C) and AIY cytoplasmic marker (P*ttx-3*::mCherry) (A', B', C') at the adult Day 1 of wild-type animals cultivated in 22˚C (A, A'), 25˚C (C, C') and *cima-1(wy84)* (B, B') animals. A", B" and C" are the corresponding merged channels. We noticed that the ASH axons extend posteriorly overlapping with AIY in zone 1 in *cima-1(wy84)* or wild-type animals cultivated in 25˚C animals. The dashed boxes correspond to zone 1 of AIY interneurons; the white arrow heads mark the ASH or AIY soma; the scale bar in (A) is 10μm and applies to the A'-C".
(TIF)

**S9 Fig. Temperature alters the synaptic subcellular specificity. (A)** A schematic diagram shows the low cultivation temperature conditions. The control group was cultivated at the constant 22˚C condition (gray line). The low temperature group was transferred from 22˚C (gray line) into 15˚C (blue line) since the parent generation (P0) young adult stage until the next generation (F1) adult Day 1 or Day 2 stage when the phenotype was scored. **(B)** Quantification of the percentage of animals with ectopic AIY synapses in the zone 1 region at 15˚C for wild-type and *cima-1(wy84)* mutants. Animals grown at 15˚C show significant less ectopic synapses than at 22˚C for both wild-type and *cima-1(wy84)*. **(C)** A schematic diagram shows the high cultivation temperature conditions (25˚C, red line) in different time windows. **(D)** Quantification of the percentage of animals with the ectopic AIY synaptic marker GFP::RAB-3 in the zone 1 region. Noted that both embryonic and larval stages are sensitive to the high temperature, the embryonic stage is more sensitive (compare window 3 and window 6). No ectopic synapses were observed when animals were treated after L4 stage (window 7). For (B) and (D), the total number of independent animals (N) and the number of biological replicates (n) are indicated in each bar for each genotype (N/n). Statistics are based on one-way ANOVA with Dunnett's test. Error bars are SEM. $^{*}P < 0.05$, $^{****}P < 0.0001$, n.s., not significant.
(TIF)

**S10 Fig. Osmotic and oxidative stresses do not affect the AIY synaptic subcellular specificity. (A)** A schematic diagram shows the time window for the sorbitol treatment. Young adults were grown on NGM agar plates containing 0mM (control, gray line), 200mM, 300mM, 400mM or 500 mM sorbitol (black line) seeded with OP50 until the next generation (F1) adult Day 1 when the phenotype was scored. **(B)** Quantification of the percentage of animals with ectopic AIY synapses in the zone 1 region under different concentration of sorbitol. The data show that the osmotic stress with the concentration of 500mM or less sorbitol has no effect on

the AIY synaptic subcellular specificity. **(C)** A schematic diagram shows time window for the oxidative stress treatment. Young adults were grown on NGM agar plates with OP50 with 0mM (control, gray line), 0.5mM, 2mM, 5mM or 10mM hydrogen peroxide (black line) in the specified time window. The phenotype of the next generation (F1) was scored at the adult Day 1 stage. **(D)** Quantification of the percentage of animals with the ectopic AIY synaptic marker GFP::RAB-3 in the zone 1 region corresponding to (C). The data show that the oxidative stress conditions do not affect the AIY synaptic subcellular specificity. For (B) and (D), the total number of independent animals (N) and the number of biological replicates (n) are indicated in each bar for each genotype (N/n). Statistics are based on one-way ANOVA with Dunnett's test. Error bars are SEM. n.s., not significant.
(TIF)

**S1 Excel. The detail information for strains used in this study.**
(XLSX)

**S2 Excel. The primer sequence information.**
(XLSX)

**S3 Excel. The archive of raw quantitative data.**
(XLSX)

**S1 Video. The AIY GCaMP fluorescent video in wild-type and Psra-6::EAT-4 transgenic animals.**
(MP4)

**S2 Video. The 3D model showed the anatomic relationship between ASH and AIY.**
(MP4)

## Acknowledgments

We thank the groups of Mei Zhen, Aravi Samuel, Jeff Lichtman, and Andrew Chisholm for generating and interpreting EM datasets for *C. elegans* connectomes, from which ASH-AIY synaptic contacts were identified. Some strains were provided by the CGC, which is funded by NIH Office of Research Infrastructure Programs (P40 OD010440). We are grateful to Dr. S. Cai, Dr. I. Mori, Dr. S. Mitani lab for strains and plasmids; Members from Shao and Colón-Ramos laboratory for their comments; and the IOBS facility core at Fudan University.

## Author Contributions

**Conceptualization:** Mengqing Wang, Zhiyong Shao.

**Data curation:** Mengqing Wang, Daniel Witvliet, Mengting Wu, Lijun Kang, Zhiyong Shao.

**Formal analysis:** Mengqing Wang, Mengting Wu, Lijun Kang, Zhiyong Shao.

**Funding acquisition:** Zhiyong Shao.

**Investigation:** Mengqing Wang, Daniel Witvliet, Mengting Wu.

**Methodology:** Mengqing Wang, Daniel Witvliet.

**Project administration:** Zhiyong Shao.

**Supervision:** Zhiyong Shao.

**Validation:** Mengqing Wang, Zhiyong Shao.

**Writing – original draft:** Mengqing Wang.

**Writing – review & editing:** Mengqing Wang, Zhiyong Shao.

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
