## [Decision Letter · Decision Letter 0]

7 Jul 2020

Dear Dr Shao,

Thank you very much for submitting your Research Article entitled 'Temperature regulates synaptic subcellular specificity through glutamatergic signaling' to PLOS Genetics. Your manuscript was fully evaluated at the editorial level and by independent peer reviewers. The reviewers appreciated the attention to an important problem, but raised some substantial concerns about the current manuscript. Based on the reviews, we will not be able to accept this version of the manuscript, but we would certainly be willing to review a much-revised version that addresses reviewer and editor comments provided below. We cannot, of course, promise publication at that time.

In addition to responding to reviewers, as editor I believe that your revised manuscript should 

1) address if high temperature is uniquely able to drive ectopic synapse formation (versus other stressors)

2) provide clarity regarding the data/results. A data table that gives results for all studies, including breaking out the individual transgenic lines reported and the p-values/statistical score, must be provided or accessible at a permanent location listed in the manuscript. For example, extrachromosomal lines were presumably scored separately and they should be reported separately in the supplementary materials, although it is certainly appropriate to merge these in the main text figures, if the lines have essentially the same impact on phenotype for each assay. 

3) explicitly state which studies were undertaken by researchers blinded as to genotype/treatment.

4) use the acknowledgement required by the CGC, with wording recommended at their website

5) corrects typos/errors in nomenclature. For example, C. elegans gene names should not be capitalized, even at the beginning of a sentence

If you decide to revise the manuscript for further consideration at PLOS Genetics, please aim to resubmit within the next 60 days, unless it will take extra time to address the concerns of the reviewers, in which case we would appreciate an expected resubmission date by email to plosgenetics@plos.org.

[LINK]

We are sorry that we cannot be more positive about your manuscript at this stage. Please do not hesitate to contact us if you have any concerns or questions.

Yours sincerely,

Anne C. Hart

Associate Editor

PLOS Genetics

Gregory Barsh

Editor-in-Chief

PLOS Genetics

Reviewer's Responses to Questions

**Comments to the Authors:**

Reviewer #1: Review of “Temperature regulates synaptic subcellular specificity through glutamatergic signaling”

This is an interesting manuscript examining the role of activity in synaptic development. The entre to the work was the identification of mutations in cima-1, a protein that appears to be a sialic acid transporter, in a screen for AIY synaptic defects. The loss of cima-1 leads to AIY neurons forming ectopic synapses in a region where they normally do not make synapses. Here the authors link that defect to glutamatergic signaling via 2 glutamate-gated chloride channels. The work is rigorous, the results novel and conclusions justified. That manipulating inhibits or promotes ectopic (or inappropriate) synapse formation is very exciting. There is an impressive amount of work that has been done very carefully and thoroughly, and will be of interest to the general field.

All of that being said, I felt there was a relative lack of consideration of the broader literature. There ares several papers examining the role of calcium channels, neurotransmitters and kinsesin-dependent transport regulation of synaptic development, none of which is referenced. One example, but not the only one, would be the 2003 Gally and Bessereau paper documenting that unc-25 mutants exhibit normal synaptogenesis, or the unc-104 mutants documenting synapses will form at cell bodies in some transport mutants. Glutamate released from glia regulate extrasynaptic glutamate receptor accumulation in many contexts, first identified in Drosophila, and is now documented in many systems. There is an interaction between glial glutamate and dopamergic neuron survival in C. elegans. These papers do nothing to diminish the novelty of the work presented here, but provide a framework for why activity, per se, is not required for normal synapse formation.

Thus, my only major concern is the lack of attention to the previous literature in the field.

I have a only a few minor things I think should be addressed before being acceptable for publication

Line 24 - “induced severely ectopic synapses” could the authors describe what makes the ectopic synapses “severe” I’m not sure I understand what that means.

Given the ~20% of wild-type animals with ectopic synapses, could the authors comment whether growth at 15 eliminates this?

Does the structure of the AIY neuron (outgrowth, branching, etc.) change appreciably when animals are reared at 25C?

In the eat-5 mutants do the GLC-3 or GLC-4 proteins accumulate in zone 1?

Reviewer #2: Neuronal activity can affect synapse formation, but the underlying molecule mechanisms are still unclear. In this manuscript, the authors use the C. elegans AIY neurons as a model to address this question. Although the observations are interesting, the main conclusions are not well supported by the data, and many results were misinterpreted.

Major issues

1) The authors started from analyzing cima-1 suppressors and showed that unc-13(lf), eat-4(lf), glc-3(lf), and glc-4(lf) suppressed cima-1 (wy84) phenotype. As wy84 is a missense mutation, the suppression of wy84 could be allele specific, and it is necessary to confirm their phenotypes using a deletion allele of cima-1.

2) Based on the suppression of cima(wy84) by eat-4(lf) and the inducement of the cima(wy84)-like phenotypes by overexpressing eat-4, the author concluded that the release of Glutamate was required for AIY synapse formation. eat-4 is essential for transport of Glu into synaptic vesicles, but eat-4(lf) also displays other notable phenotypes, such as eating disorder, defects in sensing temperature change, which may or may not associated with Glu releases from synapse terminals. To show that the cause of suppression of cima(wy84) by eat-4 is due to Glu release, the author can supply cima(wy84);eat-4 animal with Glu to examine the release of suppression, and the detail method of this treatment can be found in Leon Avery 1997 EMBO J paper (or other Avery lab’s manuscripts) .

The author also claimed that eat-4(over expression) caused “Glutamatergic neuon overactivation” and “over release of Glutamate”, which were not based on any data. EAT-4 transports Glu into synaptic vesicles, which is unlikely to activate “Glutamatergic neuon”. As the release of Glu-contained synaptic vesicles depend on neuronal activity, overexpression of eat-4 is also unlikely enough to cause ove-release of Glu. By saying that, even if overexpression of eat-4 could overload Glu into synaptic vesicles, those glutamate may not be able to release appropriately without stimulation of ASH neurons. To support the conclusion, the author need to show overexpression of eat-4 can active ASH neurons by calcium imaging, and the increase release of Glu using VGLUT-pHluorin (see Bargmann lab 2017 Elife paper).

The Glutamatergic neuronal identity of ASH is regulated by unc-42 in a very specific manner (see Hobert lab 2013 Cell paper), and the author may want to confirm the function ASH neurons by examining the role of unc-42. The function of ASH neurons can also be confirmed by killing ASH neurons using laser ablation or ced-3 overexpression.

It was motioned in the manuscript that the activation of GLC-3 GLC-4 by Glu could suppress the activity of AIY neurons and cause the phenotypes, but no evidence was shown. It could be tested by silencing AIY neurons by expression of gain of function potassium channels (such as, slo-19(gf), UNC-103(gf), see Bargmann lab 2016 Cell paper).

3) The link between environmental temperature and synaptic phenotypes is weak. Based on the suppression of “extra” synapses by eat-4 in 25C, the author concluded that the “glutamatergic activity is required for the high temperature”, again without direct evidence. To reach the conclusion, one needs to show 25C can activate ASH and increase Glu release from ASH. eat-4 has been shown to play a role in sensing temperature changes in worms, and it is possible the suppression caused by eat-4 is due to insensitive to temperature.

4) Based on data from eat-4 rescue and EM study, the author concluded that Glu released from ASH could activate GLC-3 GLC-4 in AIY neurons to regulate synapse formation. However, GLC-3 and GLC-4 are both concentrated in the AIY presynaptic buttons (to the downstream neurons), not at the post-synaptic region of ASH-AIY synapses. The question is how the Glu released from ASH neurons to activate GLC-3/GLC-4 residing outside of ASH-AIY synapses. One possibility is the “over released” Glu may diffuse to reach GLC-3/GLC-4, but this will be inconsistent with the eat-4 rescue data, in which expression of eat-4 in other glutamatergic neurons did not rescue the phenotype. Some explanation or model will be needed to facility the understanding of the data.

Reviewer #3: In this manuscript, Wang et al. show that the C. elegans interneuron AIY forms ectopic presynapses in specific genetic backgrounds or after cultivation of worms at high temperature. Wang et al. show that these ectopic presynapses are regulated by the vesicular glutamate transporter EAT-4 in ASH and postsynaptic glutamate-gated chloride channels GLC-3 and GLC-4 in AIY. The authors previously showed that presynaptic markers (RAB-3 and SYD-1) accumulate in an asynaptic region of AIY (defined as zone 1) in mutants lacking the sialin homolog CIMA-1 in a manner that is partly dependent on the VCSC glia. This study investigates whether neurons contribute to the formation of these ectopic presynapses. Wang et al. show that the ectopic RAB-3 puncta observed in cima-1 mutants are suppressed by mutants lacking the synaptic vesicle priming protein unc-13 or the vesicular glutamate transporter eat-4. In an impressive set of rescue experiments, they use a large panel of cell-specific promoters to reveal that eat-4 acts largely in the glutamatergic neuron ASH to regulate these ectopic synapses. Overexpression (OE) of eat-4 under its own promoter or promoters driving expression in ASH also result in ectopic RAB-3 puncta. A thorough analysis of potential glutamate receptors identified the glutamate-gated chloride channels GLC-3 and GLC-4 as being required for ectopic presynapse formation. Rescue experiments show that glc-3 and glc-4 function in AIY, and in a nice experiment the authors show that overexpression of glc-3 and glc-4 in AIY induces ectopic RAB-3 puncta that are dependent on eat-4, and thus likely dependent on presynaptic glutamate. Subcellular localization studies show that mCherry-tagged GLC-3 and GLC-4 appear to localize to presynaptic areas in AIY and mislocalize to zone 1 in cima-1 mutants and in eat-4(OE) animals. Wang et al. use EM and GRASP to identify previously undescribed contacts between ASH and AIY (although see comments below), providing a direct explanation for how eat-4 in ASH could regulate GLC-3 and GLC-4 in AIY. Lastly, the authors show that cultivating worms at high temperature (25 degrees C) surprisingly results in ectopic RAB-3 puncta and this effect is dependent on eat-4 and glc-3 and glc-4. The experiments are thorough and rigorous and the data are, for the most part, convincing. There are several novel findings in this manuscript including the observation that glutamate regulates the formation of ectopic presynapses in AIY via chloride-gated channels, that the glutamatergic ASH neuron potentially makes synapses with AIY, and that high cultivation temperature induces ectopic presynapses.

Major comments

1) Do the ectopic RAB-3 and SYD-1 puncta in zone 1 represent true ectopic synapses or just ectopic accumulation of presynapses without corresponding postsynapses? Either evidence from prior publications or from experiments should be provided, or statements referring to “ectopic synapses” should be toned down.

2) More background information should be included in the Introduction section about cima-1 and the the nature and function of the ectopic cholinergic synapses in AIY. For example, are the postsynaptic partners of the ectopic presynapses in zone 1 known? Is it known if the synapses are merely being misplaced from zone 1 to zone 2 with no change in the connections between AIY and its normal postsynaptic partners, or are the postsynaptic partners altered? Are there any known functional consequences of these ectopic presynapses or changes in “synaptic subcellular specificity”?

3) Although this paper focuses on ectopic presynaptic markers in zone 1, data presented in several Figures suggest that the number or intensity of RAB-3 and/or SYD-1 puncta in zone 3 are also increased in cima-1 mutants (see Figures 1D and 1J). Do cima-1 mutants have an overall increase in presynapses throughout AIY in zones 1-3? In addition, overexpression of GLC-3 and GLC-4 appear to increase RAB-3 puncta in zone 3 and this effect is not dependent on eat-4 (Figures 4D-F). The authors should comment on these effects in zone 3.

4) The synaptic connections illustrated in the EM images are not clear. (i) The electron dense presynaptic active zone structure in Figure 6A (Adult 3 bottom row) is not apparent. (ii) It is not clear that the electron dense structure marked by the arrowhead in Adult 2 (top row) is an active zone. The structure seems large and AIY does not appear to contact ASH. There is also another large dense structure nearby in the middle of the same neuron. (iii) The electron dense structure marked by the arrowhead in Adult 2 (bottom row) does not appear to contact AIY, but instead appears to make contact with another cell.

5) The presence of GRASP signal in zone 3 in the nerve ring region of AIY is concerning. The EM reconstruction data shown in Figure 6A suggest that AIY and ASH make little if any contact in the nerve ring. This data suggests that expression of the GRASP constructs in ASH and AIY leads to abnormal contact between these neurons and perhaps disorganization of other processes in the nerve ring. These non-physiological contacts detract from the use of GRASP to illustrate contact between ASH and AIY in zone 1.

6) The paper would be strengthened if the authors could show that the effects of temperature on ectopic synapses are mediated by ASH. For example, does expression of eat-4 in ASH rescue the eat-4 suppression of high temperature induced ectopic synapses? Is the ASH process displaced posteriorly alongside AIY similar to what was observed in cima-1 mutants in Figure S7?

Minor comments

1) There are many instances of incorrect grammar usage throughout the manuscript. These grammatical errors should be corrected.

2) The rationale for investigating the effects of high cultivation temperature on ectopic synapses should be explained in more depth in the Introduction. For example, is there a connection between the role of AIY in thermotaxis and the effects of high cultivation temperature?

3) Overexpression of EAT-4 is not necessarily equivalent to “activating glutamate neurons” or “increase of glutamate release.” Even if more glutamate is loaded into presynaptic vesicles due to more copies of VGLUT/synaptic vesicle, this will not lead to activation of glutamate neurons. Is there evidence in the literature that overexpression of VGLUT is sufficient to increase glutamate release? If not, the authors should provide evidence that eat-4(OE) increases glutamate release or tone down their statements equating eat-4(OE) to “activating glutamate neurons or “increase of glutamate release” throughout the manuscript.

4) In Figure 1D, zone 2 appears much smaller in cima-1 mutants. Are the presynaptic clusters spreading along the process into zone 1? If so, this information could inform the mechanism of how the ectopic synapses are forming.

5) Figure 5 shows that GLC-3 and GLC-4 colocalize with Rab-3, which is presumably on acetylcholine-containing synaptic vesicles. Are these receptors localized at presynapses in adults? And if so, the authors should comment on a physiological role for GLC-3 and GLC-4 at excitatory presynapses in the mature brain after synapses have formed. Are there examples from other systems where inhibitory receptors such as GABA receptors are localized at excitatory presynapses?

6) The authors should tone down their conclusion that GLC-3 and GLC-4 act locally in zone 1 to regulate presynaptic assembly (line 286) as GLC-3 and GLC-4 could also act in zone 3 in the nerve ring where they are also localized.

7) Zone 1 should be marked on the GRASP images in Figure 6. For example, it is not clear if the zone 1 region of AIY is visible in Figure 6E.

8) The term “N = number of animals, n = number of times scored” shown on the bar graphs should be clarified. Does this mean the same animal was scored several times and included in the data set or that N animals were scored over 6 different imaging sessions?

**Have all data underlying the figures and results presented in the manuscript been provided?**

Reviewer #1: Yes

Reviewer #2: Yes

Reviewer #3: Yes

PLOS authors have the option to publish the peer review history of their article (what does this mean?). If published, this will include your full peer review and any attached files.

Reviewer #1: No

Reviewer #2: No

Reviewer #3: No

---

## [Decision Letter · Decision Letter 1]

29 Oct 2020

Dear Dr Shao,

Thank you very much for submitting your Research Article entitled ' Temperature regulates synaptic subcellular specificity mediated by inhibitory glutamate signaling ' to PLOS Genetics. Your manuscript was fully evaluated at the editorial level and by three independent peer reviewers. The reviewers appreciated the attention to an important topic and make clear that you have addressed all of their scientific concerns. But in their comments to the journal, they identified one aspect of the manuscript that should be improved,- specifically correction of errors, typos, and/or English language usage. 

Therefore, we ask you to proofread and correct the manuscript and resubmit the revised version, without changing scientific content. To be clear, I do not expect to send the manuscript to the reviewers again. Instead, I will evaluate the manuscript you revise.

[LINK]

Yours sincerely,

Anne C. Hart

Associate Editor

PLOS Genetics

Gregory Barsh

Editor-in-Chief

PLOS Genetics

Reviewer's Responses to Questions

**Comments to the Authors:**

Reviewer #1: The author's have satisfied my concerns with their thorough response. I approve of this manuscript for publication.

Reviewer #2: They have addressed all my questions regarding experiments. It is a very interesting study, but the way the authors wrote the manuscript makes it difficult to follow. I will suggest a minor revision with a some corrections and clarifications on writing before acceptance for publication.

Reviewer #3: The authors have added a significant number of informative experiments in the revised version of the manuscript including cell ablation of ASH, analysis of other stresses and measurement of GCaMP in AIY and VGLUT-pHluorin in ASH. All my concerns have been addressed by the authors.

**Have all data underlying the figures and results presented in the manuscript been provided?**

Reviewer #1: Yes

Reviewer #2: Yes

Reviewer #3: Yes

PLOS authors have the option to publish the peer review history of their article (what does this mean?). If published, this will include your full peer review and any attached files.

Reviewer #1: No

Reviewer #2: No

Reviewer #3: No

---

## [Editor Report · Decision Letter 2]

5 Dec 2020

Dear Dr Shao,

We are pleased to inform you that your manuscript entitled "Temperature regulates synaptic subcellular specificity mediated by inhibitory glutamate signaling" has been editorially accepted for publication in PLOS Genetics. Congratulations!

Yours sincerely,

Anne C. Hart

Associate Editor

PLOS Genetics

Gregory Barsh

Editor-in-Chief

PLOS Genetics

Comments from the reviewers (if applicable):

**Data Deposition**

http://datadryad.org/submit?journalID=pgenetics&manu=PGENETICS-D-20-00854R2

**Press Queries**

---

## [Editor Report · Acceptance letter]

5 Jan 2021

PGENETICS-D-20-00854R2 

 Temperature regulates synaptic subcellular specificity mediated by inhibitory glutamate signaling 

Dear Dr Shao, 

We are pleased to inform you that your manuscript entitled " Temperature regulates synaptic subcellular specificity mediated by inhibitory glutamate signaling " has been formally accepted for publication in PLOS Genetics! Your manuscript is now with our production department and you will be notified of the publication date in due course.

With kind regards,

Livia Horvath

PLOS Genetics

On behalf of:
